# Transcriptional alterations of peanut root during interaction with growth-promoting *Tsukamurella tyrosinosolvens* strain P9

**Xue Bai, Yujie Han, Lizhen Han** [ID]*

College of Life Sciences, Key Laboratory of Plant Resource Conservation and Germplasm Innovation in Mountainous Region (Ministry of Education), Institute of Agro-Bioengineering, Guizhou University, Guiyang, Guizhou, China

* lzhan1@gzu.edu.cn

**Data Availability Statement:** All relevant data are available from the NCBI database (accession numbers PRJNA991079) and Supporting Information files.

## Abstract

The plant growth-promoting rhizobacterium *Tsukamurella tyrosinosolvens* P9 can improve peanut growth. In this study, a co-culture system of strain P9 and peanut was established to analyze the transcriptome of peanut roots interacting with P9 for 24 and 72 h. During the early stage of co-culturing, genes related to mitogen-activated protein kinase (MAPK) and $Ca^{2+}$ signal transduction, ethylene synthesis, and cell wall pectin degradation were induced, and the up-regulation of phenylpropanoid derivative, flavonoid, and isoflavone synthesis enhanced the defense response of peanut. The enhanced expression of genes associated with photosynthesis and carbon fixation, circadian rhythm regulation, indoleacetic acid (IAA) synthesis, and cytokinin decomposition promoted root growth and development. At the late stage of co-culturing, ethylene synthesis was reduced, whereas $Ca^{2+}$ signal transduction, isoquinoline alkaloid synthesis, and ascorbate and aldarate metabolism were up-regulated, thereby maintaining root ROS homeostasis. Sugar decomposition and oxidative phosphorylation and nitrogen and fatty acid metabolism were induced, and peanut growth was significantly promoted. Finally, the gene expression of seedlings inoculated with strain P9 exhibited temporal differences. The results of our study, which explored transcriptional alterations of peanut root during interacting with P9, provide a basis for elucidating the growth-promoting mechanism of this bacterial strain in peanut.

## Introduction

Peanut (*Arachis hypogaea* L.) is an important economic and oil crop that is widely grown in many countries [1]. Although chemical fertilizers can boost crop yields, their application can have a detrimental impact on the environment. The potential of plant growth-promoting rhizobacteria (PGPR) as biofertilizers to enhance crop growth and improve yields is receiving increasing attention [2]. Most reported PGPR belong to genera such as *Agrobacterium*, *Arthrobacter*, *Azospirillum*, *Azotobacter*, *Bacillus*, *Bradyrhizobium*, *Burkholderia*, *Pseudomonas*, and *Rhizobium*. Research has demonstrated that PGPR can enhance plant growth through a variety

**Funding:** This work was supported by the the National Natural Science Foundation of China (32060028). In all of authors, Lizhen Han is the corresponding author, Xue Bai and Yujie Han are Lizhen Han's graduate students. Lizhen Han designed the experiments, Xue Bai investigated this study and wrote the manuscript, the two students performed the experiments.

**Competing interests:** The authors declare that they have no competing interest.

of probiotic mechanisms, including phosphate solubilization, nitrogen fixation, siderophore production, indoleacetic acid (IAA) synthesis, and secretion of 1-aminocyclopropane-1-carboxylic acid (ACC) deaminase [3, 4]. In contrast, how plants respond to PGPR is less well studied. Studies have shown that inoculation of *Azospirillum* sp. into rice and wheat induces the expression of cytokinin regulatory genes, amino acid metabolism-related pathways, and antioxidant synthesis [5, 6]. *Burkholderia cepacia* MYSP113 has various effects on gene expression in sugarcane roots, thereby affecting processes such as amino acid biosynthesis and metabolism, phytohormone signaling, plant–pathogen interactions, carbon metabolism, and fatty acid degradation [7]. The response mechanism of plants to PGPR is an obvious and direct explanation for the growth-promoting effect of PGPR.

Actinomycetes are one of the important components of soil microorganisms, known for producing diverse antibacterial and antitumor active substances. In recent years, there have been reports of *Streptomyces* sp. strains improving the growth of crops such as wheat, tomato, rice [8–10]. However, there is still little research and understanding of rare actinobacteria [11], and their growth-promoting properties have seldom been reported [12]. In our previous study, we isolated and screened a rare actinomycete, *Tsukamurella tyrosinosolvens* P9, that exhibits strong phosphate solubilization, IAA secretion, and siderophore production for the first time. This strain can not only effectively colonize the root tissues of peanut, but also can promote peanut growth by affecting the microbial diversity of rhizosphere soil [13, 14]. Nevertheless, the response mechanism of peanut seedlings to strain P9 over time is still unknown. In this study, we established a co-culture system between strain P9 and peanuts and conducted a transcriptome analysis of peanut roots at two different time points to study the gene expression of peanut roots after inoculation with the growth-promoting P9 strain. Our aim was to provide direct evidence for the molecular mechanism of strain P9 in promoting peanut growth, and to provide a theoretical basis for using *Tsukamurella tyrosinosolvens* strain P9 to the field level.

## Materials and methods

### Bacterial strain and plant material

In this study, we used seeds of Silihong (Virginia runner type), an early maturing, local peanut variety with a dark red seed coat. The bacterial strain of *Tsukamurella tyrosinosolvens* was P9, which was isolated in our laboratory and stored at the China Typical Culture Preservation Center (strain preservation number CCTCC AA 2020052) and in our laboratory as well.

### Preparation of bacterial solution

An appropriate amount of strain P9 was added to 50 mL LB liquid medium and then incubated at 30°C with shaking at 180 rpm for 16 h to activate. The activated cultures were transferred to LB liquid medium, incubated for 24 h, and then centrifuged at 5000 $\times g$ for 10 min. The supernatant was discarded, and the bacterial precipitate was rinsed twice with 5 mL sterile water. Finally, a bacterial suspension ($10^8$ cfu/mL) was prepared in half-strength MS liquid medium.

### Construction of the strain-P9–peanut-seedling co-culture system and preparation of transcriptome samples

Peanut seeds were disinfected with 20% $H_2O_2$ solution for 20 min, washed five times with sterile water, and soaked for 8 h. The seeds were placed in a Petri dish covered with moist sterile filter paper and then stored in a light incubator (16-h/8-h photoperiod) at 28°C for 3 days until germination. Germinated peanut seedlings with consistent growth were selected and

transplanted into a sterile culture bottle containing pearl cotton and 195 mL half-strength MS liquid medium. After the leaves had unfolded, one set of peanut seedlings were inoculated with 5 mL of the P9 bacterial suspension ($1 \times 10^8$ cfu/mL) as the treatment group and cultured in a 28˚C light incubator, while the same volume of half-strength MS medium was added to other seedlings as the control group. Treatment and control groups were further divided as follows: CK1 (24-h control group), CK2 (72-h control group), T1-P9 (treatment group co-cultured with P9 for 24 h), and T2-P9 (treatment group co-cultured with P9 for 72 h). After 24 and 72 h of co-culturing, root samples of the different treatment and control groups were frozen in liquid nitrogen for transcriptome sequencing analysis, and seedling growth indicators and chlorophyll contents were measured as well.

## Total RNA extraction and transcriptome analysis of peanut roots

Total RNA was extracted from peanut roots using a TRNzol-A total RNA isolation kit (TIAN-GEN, Beijing, China). The RNA was checked for quality and quantity using a Nano 6000 Assay kit and a 2100 Bioanalyzer system (Agilent Technologies, Palo Alto, CA, USA). After quantification of RNA with a Qubit RNA Assay kit and a Qubit2.0 fluorometer (Life Technologies, CA, USA), a cDNA library was constructed using a NEBNext Ultra RNA Library Prep Kit for Illumina kit (NEB, USA) and then sequenced on the Illumina HiSeqTM2500 platform (Novogene 6000, Beijing, China).

Raw reads were subjected to Trimmomatic software (v0.40) to remove adapters, poly-N-containing reads, and low-quality reads, Q20, Q30, and GC contents of clean data were calculated. The genome of *Arachis hypogaea* (GenBank accession number GCF_003086295.2_ar-ahy.Tifrunner.gnml.KYV3) was used as a reference sequence [15], and the clean data were mapped against the reference genome using the software Hisat2 (v2.0.1). Sequences with total mapped reads or fragments larger than 70% of the data were selected for subsequent analysis. Gene expression was calculated using the fragments per kilobase of exon model per million mapped fragments (FPKM) method. The threshold FPKM >1 was used as the criterion for identification of DEGs. DEGs were screened using edgeR software (selection criteria: $P < 0.05$ and |log2(foldchange)| > 1). and their gene ontology (GO) and Kyoto Encyclopedia of Genes and Genomes (KEGG) pathways were analyzed using clusterProfile R package. RNA-seq raw data has been uploaded to the NCBI database (Accession number PRJNA991079, https://www.ncbi.nlm.nih.gov/bioproject/PRJNA991079).

## Quantitative real-time PCR verification

Using the results of the KEGG pathway DEG analysis of peanut roots, we selected 24 candidate genes at the two stages (24 and 72 h after inoculation) for quantitative real-time PCR (RT-qPCR) verification. Primers for RT-qPCR were designed using Primer-BLAST (S1 Table) and then synthesized by Shanghai Bioengineering Co. The actin gene was used as an internal control. After RT-qPCR amplification, relative gene expression levels were calculated using the $2^{-\Delta\Delta Ct}$ method based on the $C_t$ values of each sample [16].

## Data processing and analysis

Data were analyzed for significant differences using the independent sample *t*-test in IBM SPSS Statistics 25.0. The following criteria were used to indicate significant and very significant differences, respectively: $P < 0.05$ and $P < 0.01$. Bar charts were constructed using Microsoft Excel 2010.

## Results

### Establishment of a P9–peanut-seedling co-culture system

A hydroponics-based co-culture interaction system was established between *Tsukamurella tyrosinosolvens* P9 and peanut seedlings. As shown in Figs 1 and 2, the root length, lateral root number, and chlorophyll content of peanut seedlings at the initial stage of co-culturing with strain P9 (24 h after inoculation) were increased by 8.34%, 6.67%, and 5.26%, respectively, compared with those in control seedlings. At the late stage of co-culturing (72 h after inoculation), chlorophyll content was increased by 17.65%, plant height was significantly different ($P < 0.05$), and fresh weight, root length, root weight, and lateral root number were highly significantly different ($P < 0.01$) compared with those in the control. Seedling roots were more abundant and thicker, and stem and leaf growth was also significantly better than that of the control. These results demonstrate that strain P9 had positive effects on peanut growth 24 h after inoculation and that the effect was more significant 72 h after inoculation. These two co-culturing time points were therefore used for the preparation of root transcriptome samples.

### Peanut root transcriptome sequencing and quality analysis

As shown in S2 Table, transcriptome sequencing of the 24-h peanut root sample (CK1) yielded an average of 46,253,588 and 44,135,290 raw and clean reads, respectively, while an average of 43,823,152.67 raw reads and 41,972,738 clean reads were generated from the 72-h peanut root sample (CK2). In regard to P9-strain-treated peanut groups, the average number of raw and

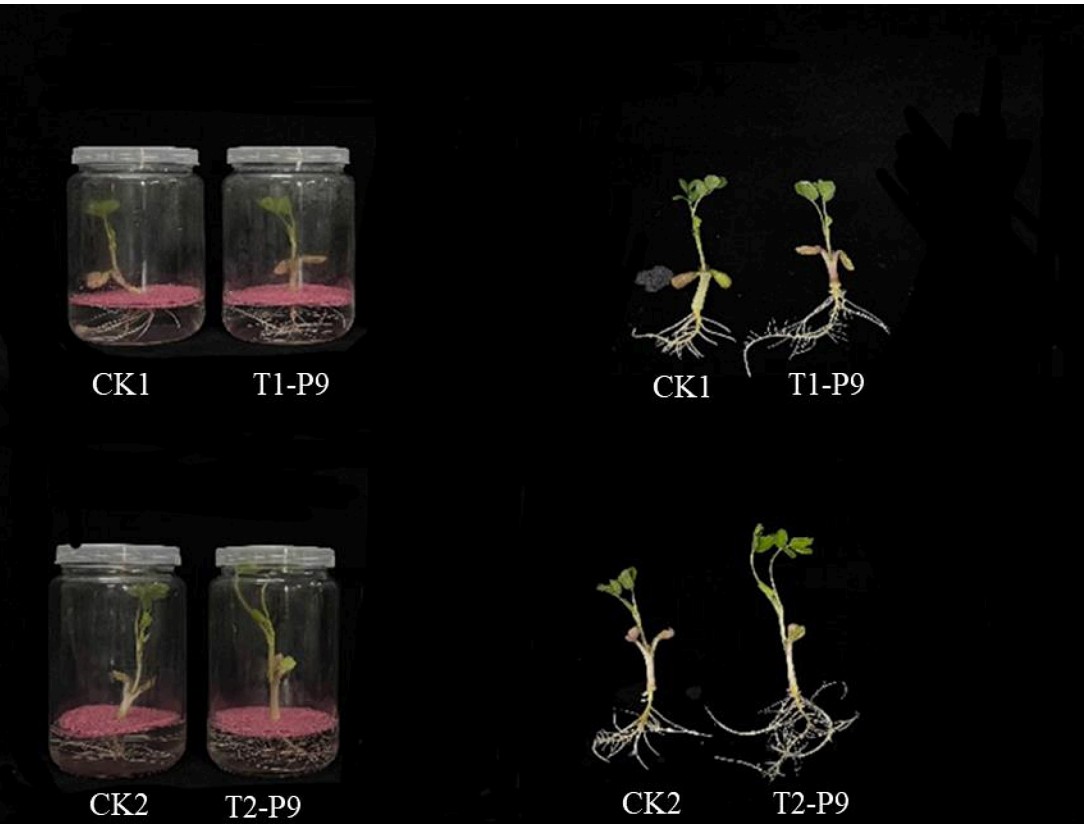

**Fig 1. Growth status of peanut inoculated with *Tsukamurella tyrosinosolvens* strain P9 after 24 and 72 h.**

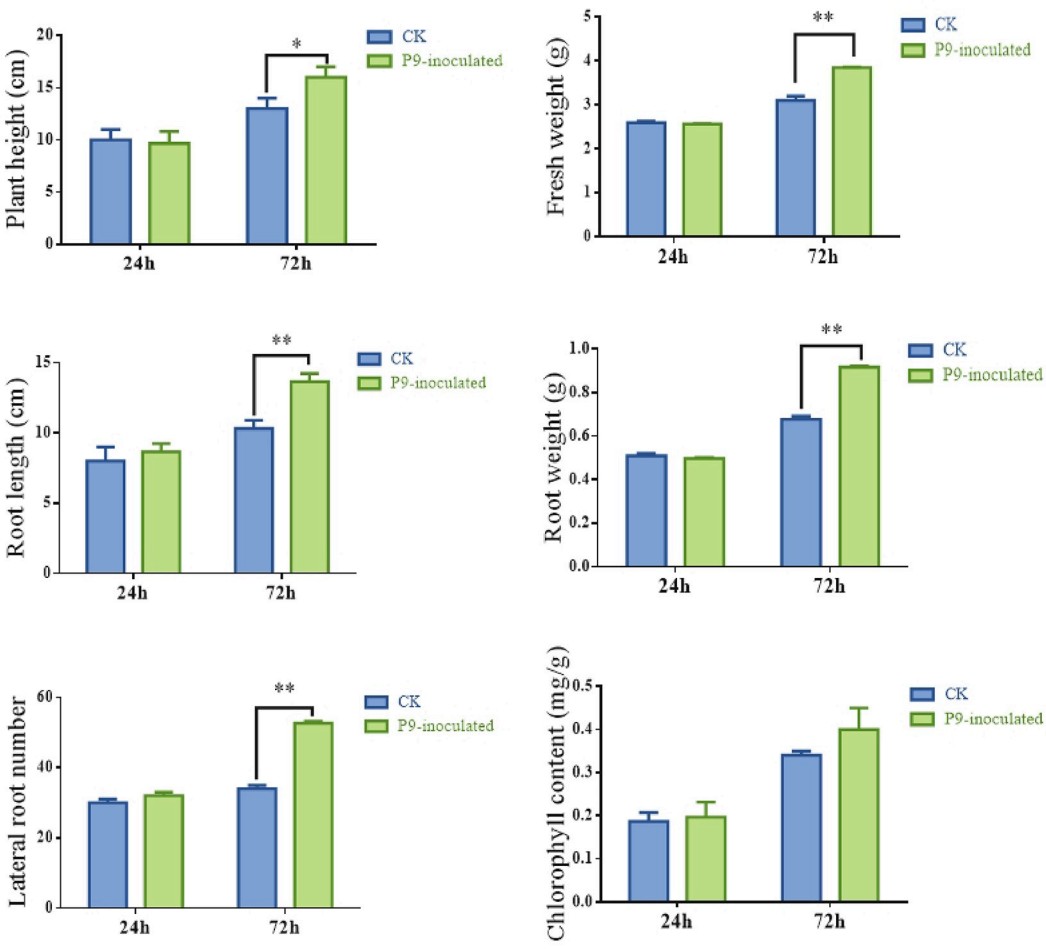

**Fig 2. Growth indices and chlorophyll contents of peanut seedlings co-cultured with *Tsukamurella tyrosinosolvens* P9.** Error bars show standard deviation (SD) of the mean of three replicates (*, *P* < 0.05; **, *P* < 0.01).

clean reads obtained from roots inoculated with strain P9 for 24 h (T1-P9) was 44,791,624 and 42,654,016, respectively, whereas the average from roots inoculated with P9 for 72 h (T2-P9) was 45,770,172 and 43,806,935.33, respectively. Q30 values of the data from different treatment groups were all higher than 94.00%, and the percentage of sample sequences matching the reference genome was higher than 94.34%. These results confirm the quality of the data and the suitability of the selected reference genome, thus validating their use in the transcriptome analysis.

### Differentially expressed gene (DEG) and KEGG enrichment analyses of peanut roots in response to strain P9

We observed differences in the gene expression of peanut roots co-cultured with strain P9 (S1 Fig). Compared with the genes in the control group at the same time point, 2,271 genes were differentially expressed in peanut roots co-cultured with strain P9 for 24 h (total of 1,559 up-regulated genes and 712 down-regulated genes), and 1,479 genes were differentially expressed in roots after 72 h of co-culturing with strain P9 (total of 912 up-regulated genes and 567 down-regulated genes). In addition, at 24 h post-inoculation, there were 2,206 DEGs shared between treatments, with 1,555 ~~were~~ up-regulated and 651 down-regulated. At 72 h post-

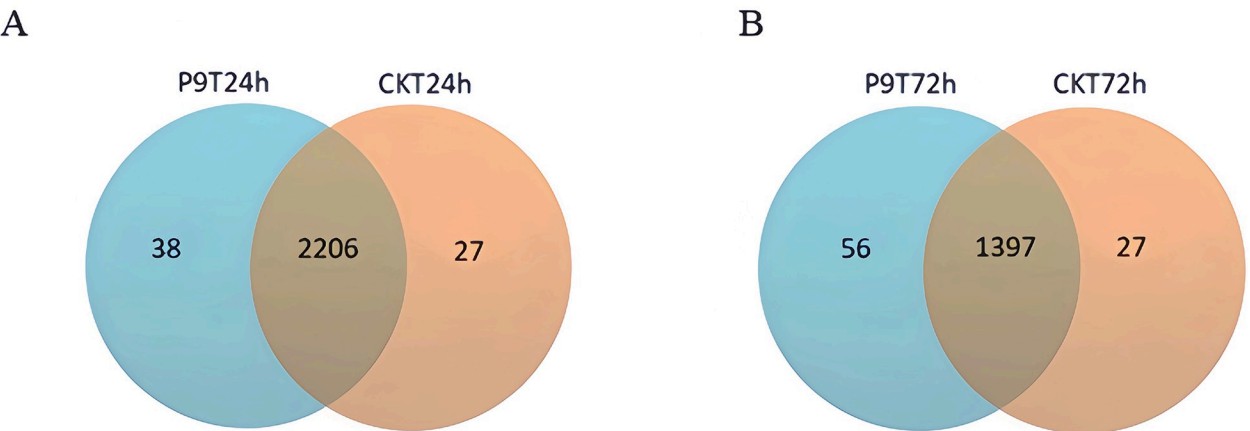

**Fig 3.** Venn diagram of differential gene volcanoes in peanut roots inoculated with *Tsukamurella tyrosinosolvens* P9 for 24 h (left) or 72 h (right). The non-overlapping region was the number of unique DEGs of peanut inoculated with P9 strain, and the overlapping region was the number of common DEGs.

inoculation, there were 1,397 DEGs, with 607 up-regulated and 701 down-regulated, respectively. Additionally, there were 38 and 56 unique genes identified at 24 and 72 hours post-inoculation, respectively (Fig 3). At 24 h after inoculation, the roots expressed 38 unique genes, mostly related to transcription factors. At 72 h after inoculation, the roots expressed 56 unique genes involved in multiple metabolic pathways, such as starch and sucrose metabolism, amino sugar and nucleotide sugar metabolism, pyruvate metabolism, oxidative phosphorylation, glycine, serine and threonine metabolism, pentose and glucuronate interconversions. These results suggest that inoculation with strain P9 affects the gene expression of peanut seedlings and that this regulation is characterized by temporal differences. The heatmap analysis revealed differences in gene expressions of peanut roots after inoculation with the P9 strain at different time points, especially at 24 and 72 hours (Fig 4).

KEGG enrichment analysis (Fig 5) revealed that genes differentially expressed at the early stage of co-culturing were associated with 111 KEGG pathways. Among these pathways, 14 metabolic pathways were significantly enriched in DEGs: flavonoid biosynthesis; circadian rhythm-plant; isoflavone biosynthesis; phenylpropanoid biosynthesis; linoleic acid metabolism; zeatin biosynthesis; biosynthesis of various secondary metabolites-Part 2; pentose and glucuronate interconversions; cysteine and methionine metabolism; carbon fixation in photosynthetic organisms; betalain biosynthesis; cutin, suberin, and wax biosynthesis; mitogen-activated protein kinase (MAPK) signaling pathway-plant; and photosynthesis. At the late stage of co-culturing, 92 metabolic pathways were enriched in DEGs. The following 10 pathways were significantly enriched: pentose and glucuronate interconversions; glycine, serine, and threonine metabolism; zeatin biosynthesis; nitrogen metabolism; alpha-linolenic acid metabolism; ascorbate and aldarate metabolism; glycerolipid metabolism; valine, leucine, and isoleucine biosynthesis; ABC transport system; and isoquinoline alkaloid biosynthesis. Only two pathways, namely, pentose and glucuronate interconversion and zeatin biosynthesis, were significantly enriched during both stages. Furthermore, the metabolic pathways significantly enriched in DEGs in peanut roots at the early stage of co-culturing were mainly associated with flavonoid, isoflavone, and phenylpropanoid synthesis, photosynthesis, and defense responses, whereas those significantly enriched at the late stage of co-culturing were mainly related to nutrient metabolism and energy production. These results indicate that the response

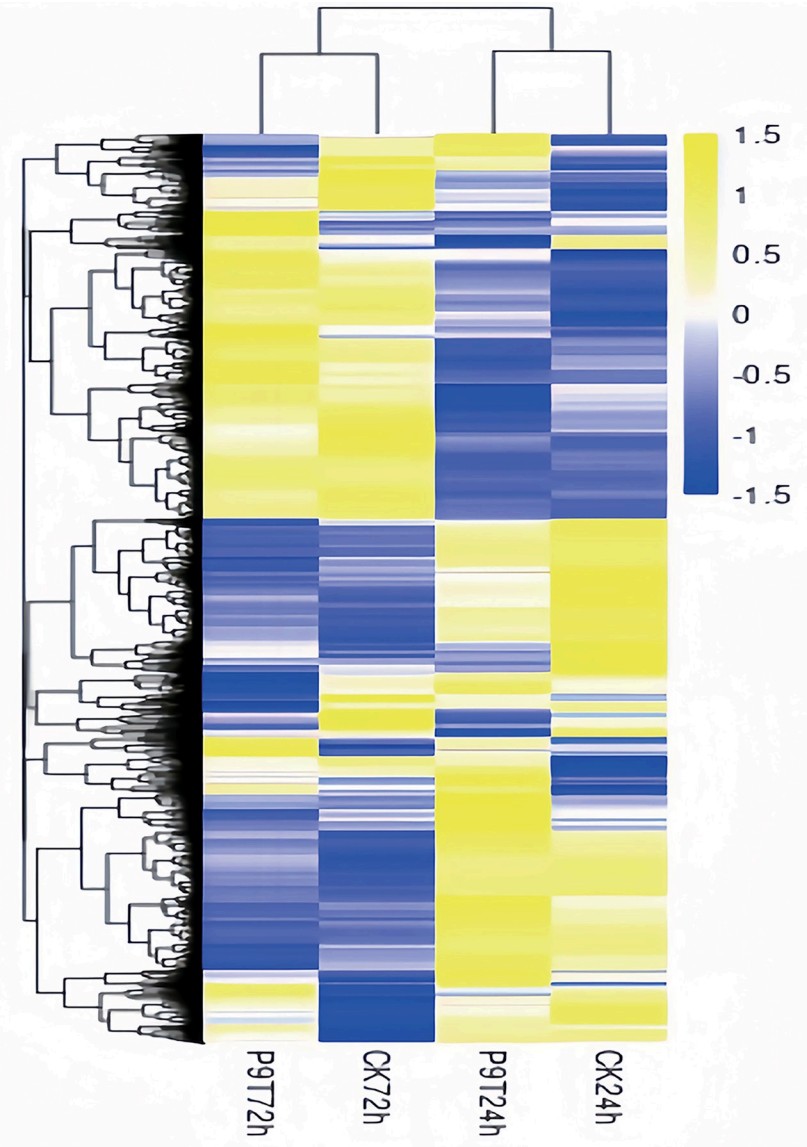

**Fig 4. Hierarchical clustering and heatmap of differentially expressed genes in peanut roots inoculated with** *Tsukamurella tyrosinosolvens* **P9 for 24 h and 72 h.** The figure displays gene expression values for this group of samples.

mechanisms and metabolic pathways of peanut seedlings interacting with strain P9 were different between the two time periods.

## Functional analysis of DEGs

We analyzed DEGs and their functions and found that inoculation with strain P9 significantly affected the transcriptome of peanut roots. Genes differentially expressed in response to strain P9 were mainly related to signal transduction, cell wall modification, plant hormone synthesis, substance and energy metabolism, and plant defense stress, but which genes were differentially expressed and which KEGG pathways were enriched in peanut after inoculation with strain P9 varied according to co-culturing stage.

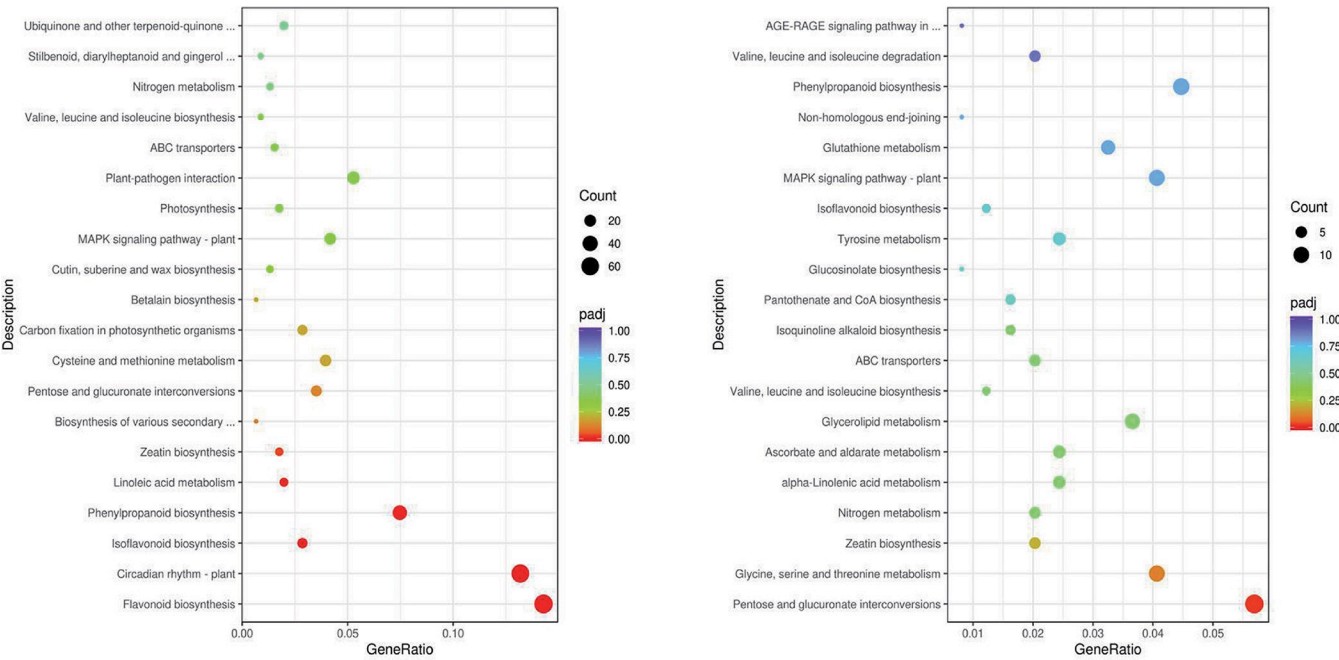

**Fig 5.** KEGG pathways enriched in DEGs in peanut inoculated with *Tsukamurella tyrosinosolvens* P9 at 24 h (left) and 72 h (right).

**Effects of strain P9 on the expression of signal transduction-related genes in peanut seedling roots.** The Co-culturing with strain P9 affected the signal transduction pathway of peanut seedlings (S3 Table). The MAPK signaling pathway, one of the most important signal transduction systems in organisms, is also closely related to the plant–pathogen interaction pathway [17]. During the early stage of co-culturing (24 h after inoculation), the expressions of LRR receptor-like serine/threonine-protein kinase FLS2 (*FLS2*) and *WRKY* transcription factor 33 (*WRKY33*) were induced, and ethylene response factor 1 (*ERF1*) and basic endochitinase B (*ChiB*) genes were significantly up-regulated–by up to 4.46- and 4.10-fold, respectively. The increased expression of these genes induced the rapid defense response of the seedlings; at the same time, the up-regulation of respiratory burst oxidase (Rboh) helped maintain the homeostasis of intracellular reactive oxygen species (ROS). $Ca^{2+}$, an important intracellular second messenger, was quickly transported and transmitted by the slight up-regulation of cyclic nucleotide gated channel CNGCs and calcium-binding protein CML (CaMCML), additionally, WRKY2 was up-regulated by 3.33-fold, all inducing seedling defense responses. At the late stage of co-culturing (72 h after inoculation), the *FLS2* gene was upregulated (up to 4.47-fold), *Rboh* and *CaMCML* genes were significantly up-regulated (up to 6.12-fold and 7.97-fold, respectively), and the *WRKY33* gene was down-regulated (3.78-fold), but the expression of the *ERF1* gene did not differ from that of the control. These results indicate that the defense response induced by ethylene was lower than that at the early stage, whereas the regulation of peanut ROS homeostasis was stronger.

**Effect of strain P9 on the expression of pectin degradation-related genes in root cell walls.** During both co-culturing stages, pentose and glucuronate interconversion pathways (S4 Table), which are related to the synthesis and degradation of cell wall components, were significantly enriched in DEGs in peanut. At the early stage of co-culturing, nine pectinesterase (*PE*) genes and two pectate lyase (*PL*) genes were up-regulated, which would have catalyzed the decomposition of pectin into more highly unsaturated digalacturonate. At the later co-

culturing stage, the up-regulation (up to 5.02-fold) of five *PE* genes increased the production of polygalacturonic acid via pectin decomposition.

**Effect of strain P9 on hormone synthesis in roots.** Inoculation with strain P9 affected the synthesis of various hormones in peanut seedlings (S5 Table). The zeatin synthesis pathway was significantly enriched in DEGs during both co-culturing periods. Five cytokinin dehydrogenase (*CKX*) genes were up-regulated at the early stage of co-culturing (up to 1.50-fold). At the late stage of co-culturing, the gene encoding UDP-glucosyltransferase 73C catalyzed *DZ* (dihydrozeatin) was down-regulated 5.28-fold, although the *CKX* gene was more significantly up-regulated (up to 6.64-fold) and cytokinin trans-hydroxylase (*CYP735A*) was significantly up-regulated, the decomposition of cytokinin was still generally up-regulated.

Co-culturing with strain P9 also affected the tryptophan metabolic pathway of peanut. During the early stage of co-culturing, the two aldehyde dehydrogenase (*ALDH*) genes catalyzing the conversion of 3-indole acetaldehyde into indoleacetic acid (IAA) were up-regulated (up to 4.32-fold), and indole-3-pyruvate monooxygenase catalyzing the conversion of indole-pyruvate into IAA was slightly up-regulated, all of which increased the synthesis of IAA. At the late stage of co-culturing, the gene encoding indole-3-pyruvate monooxygenase was down-regulated 5.90-fold.

Cysteine and methionine metabolic pathways are closely related to ethylene synthesis [18]. At the early stage of co-culturing, the up-regulation of S-adenosylmethionine synthetase (up to 3.13-fold), 1-aminocyclopropane-1-carboxylate synthase (*ACS*) (up to 5.16-fold), and aminocyclopropanecarboxylate oxidase (*ACO*) boosted the production of ethylene from methionine. At the late stage of co-culturing, the induced expression of S-adenosylmethionine synthetase and DNA (cytosine-5)-methyltransferase 1 increased the production of S-adenosyl L-homocysteine catalyzed from methionine. At the same time, the *ACO* gene was down-regulated, which resulted in a decrease in ethylene production.

In regard to the diterpenoid biosynthesis pathway, the slight down-regulation of the gene encoding ent-kaurenoic acid monooxygenase led to reduced *GA*12 content at the early stage. During the late stage of interaction, gibberellin 2 beta-dioxygenase, which catalyzes the conversion of different types of gibberellin (*GA*) to its metabolite form, was down-regulated 2.02-fold.

**Effects of strain P9 on the expression of lipid metabolism-related genes in roots.** Inoculation with strain P9 affected the glyceride and glycerophospholipid metabolic pathways of peanut (S6 Table). At 24 h of co-culturing, genes encoding phosphatidate phosphatase LPIN, phospholipase C, and phospholipid:diacylglycerol acyltransferase (*PDAT*) were all up-regulated, which increased the production of phosphatidyl-ethanolamine and triacylglycerol, and the phosphoethanolamine N-methyltransferase gene was 3.75-fold up-regulated, thereby increasing the production of phosphocholine. When peanut was co-cultured with P9 for 72 h, the D-glycerate 3-kinase (*GLYK*) gene was significantly up-regulated (up to 9.94-fold), which enhanced the phosphorylation of glyceric acid that then entered the glycolysis pathway. Up-regulation of glycerol-3-phosphate acyltransferase (*GPAT*) and phospholipase D1/2 greatly increased the production of 1,2-diacyl-sn-glycerol-3-P, and the significant up-regulation of lysophospholipase II and choline-phosphate cytidyltransferase (8.48-fold) boosted phosphatidylcholine synthesis.

When peanut was co-cultured with strain P9 for 24 h, multiple fatty acid metabolism-related pathways were induced. The gene encoding acetyl-CoA acyltransferase was up-regulated, and the production of acetyl-CoA increased. Furthermore, the up-regulation of acetyl-CoA carboxylase1, 3-oxoacyl-(acyl-carrier-protein) synthase III (*FabH*), and long-chain acyl-CoA synthetase (*ACSL*) catalyzed the synthesis of more long-chain acyl-CoA, while the induced expression of acyl-coenzyme A thioseterase 1/2/4 increased the synthesis of

unsaturated fatty acids (such as linoleic acid and α-linolenic acid) and long-chain fatty acids. At the later stage of co-culturing, enoyl-CoA hydratase (*ECH*) and *FabH* genes were down-regulated (−9.47-fold).

Strain P9 also affected linoleic acid and α-linolenic acid metabolic pathways. During the early stage of co-culturing, up-regulation of the lipoxygenase (*LOX*) gene accelerated the decomposition of linoleic acid and α-linolenic acid. The up-regulation of genes encoding hydroperoxide dehydratase, 12-oxophytodienoic acid reductase (*OPR3*), and acetyl-CoA acyl-transferase 1 produced more JA-CoA, thus providing more precursors for jasmonic acid synthesis. At the late stage of co-culturing, the gene encoding fatty acid alpha-dioxygenase (*DOX*) was up-regulated up to 5.74-fold, and the synthesis of heptadecatrienoic acid increased. Although the enoyl-CoA hydratase/3-hydroxyacyl-CoA dehydrogenase (*MFP2*) gene was slightly down-regulated, acyl-CoA oxidase (*ACOX*) was up-regulated and the *OPR3* gene was up-regulated more than at the early stage, thus increasing the production of JA-CoA.

**Effects of strain P9 on the expression of peanut genes related to photosynthesis, carbon and nitrogen metabolism, and transport.** At the early stage of co-culturing with strain P9, metabolic pathways associated with photosynthesis, carbon fixation, and circadian rhythm were significantly enriched in peanut seedlings, antenna proteins protein was induced, too (S6 Table). Regarding the photosynthesis pathway and antenna proteins pathways, the following proteins were upregulated: photosystem II oxygen-evolving enhancer proteins PsbO, PsbP, and PsbQ; photosystem I subunit II PSaD; cytochrome b6-f complex iron-sulfur subunit PetC; photosynthetic electron transport ferredoxin PetF; and the antenna proteins light-harvesting complex II chlorophyll a/b binding proteins Lhcb1 and Lhcb3. Their expression increased by 1.07 to 2.70 fold, which enhanced the photosynthesis process. Malate dehydrogenase, ribose 5-phosphate isomerase A (*rpiA*), phosphoribulokinase (*PRK*), ribulose-bisphosphate carboxyl-ase, and glyceraldehyde-3-phosphate dehydrogenase (*GAPDH*) in the carbon fixation pathway were also induced. At 72 h after inoculation, the malate dehydrogenase gene involved in carbon fixation, which catalyzes the decarboxylation of malate to pyruvate, was up-regulated 6.08-fold, which greatly increased the amount of $CO_2$.

Multiple sugar metabolic pathways in peanut were also affected by strain P9. In the early stage of co-culturing, the up-regulated expression of endoglucanase, beta-glucosidase, and chitinase (*ChiC*) led to a significant increase in glucose produced by cellulose and chitin decomposition. The up-regulation of hexokinase (*HK*), glucose-6-phosphate 1-epimerase, 6-phosphofructokinase 1 (*PFK*), and enolase accelerated glycolysis metabolism and increased phosphoenolpyruvate production. Pyruvate was catalyzed to produce more ethanol and acetic acid by the up-regulation of pyruvate decarboxylase (*PDC*), alcohol dehydrogenase (*ADH*), and aldehyde dehydrogenase (*ALDH*). In contrast, the genes encoding $H^+$-transporting ATPase, F-type $H^+$-transporting ATPase subunit gamma, and *Ndufsl* were down-regulated, thereby decreasing the amount of ATP produced by oxidation. In addition, up-regulated expression of starch synthase produced more amylose. In the late stage of inoculation, the significant up-regulation of beta-fructofuranosidase (up to 5.47-fold), *ChiC* (up to 4.62-fold), glycogen phosphorylase (*GP*, 11.37-fold), glucose-1-phosphate adenyltransferase (6.37-fold), and UTP-glucose-1-phosphate uridylyltransferase (5.63-fold) accelerated the decomposition of sucrose, chitin, and starch. Glucuronokinase, which catalyzes the phosphorylation of glucu-ronic acid, was significantly up-regulated (6.82-fold), and carbohydrate catabolism was signifi-cantly higher than in the early stage of co-culturing. Moreover, the significant up-regulation of D-lactate dehydratase (8.35-fold), malate dehydrogenase (6.08-fold), and pyruvate decarboxyl-ase (7.92-fold) genes improved the production of lactic acid, pyruvic acid, and acetaldehyde, and the amount of $CO_2$ produced increased significantly. In addition, the genes encoding $H^+$-transporting ATPase, NdhK, and Ndufb7 protein were up-regulated 3.01-, 4.55-, and

6.01-fold, respectively, and the amount of ATP produced by oxidative phosphorylation was significantly increased.

Nitrogen utilization is essential for plant growth and development. The pathway of nitrogen metabolism was significantly enriched after 72 hours of co-culturing with strain P9. In the early stage of co-culturing, nitrate reductase (*NR*) was up-regulated and nitrate assimilation was enhanced. With the extension of the co-culturing time, NR and nitrate/nitrite transporter (*NRT*) were significantly up-regulated, the nitrogen metabolism of peanut was improved, and plant nitrogen uptake and absorption was significantly increased. In addition, ATP-binding cassette was significantly up-regulated at 72 h, and *ABCB1* and *ABCB10* were up-regulated 4.27- and 9.34-fold, respectively.

**Effects of strain P9 on the expression of defense- and stress-related genes in roots.** Co-culturing with strain P9 affected multiple metabolic pathways related to resistance to biotic or abiotic stresses in peanut seedlings (S7 Table). At the 24-hour stage of co-culturing, there was a significant up-regulation in the biosynthesis of flavonoid, and isoflavonoid, phenylpropanoids, and various secondary metabolites. At the 72-h co-culturing stage, metabolic pathways such as glycine, serine, and threonine metabolism; ascorbate and aldarate metabolism; and isoquinoline alkaloid biosynthesis were significantly enriched.

During the early stage of inoculation, genes encoding trans-cinnamate 4-monooxygenase (*CYP73A*), ferulate-5-hydroxylase (*F5H*), cinnamyl-alcohol dehydrogenase (*CAD*), peroxidase, and coniferyl alcohol glucosyltransferase (*CAGT*) in the phenylpropanoid synthesis pathway were up-regulated, which promoted the synthesis of lignin and other bioactive substances. Genes encoding chalcone synthase CHS, naringenin 3-dioxygenase, and trans-cinnamate 4-monooxygenase were significantly up-regulated, and the synthesis of flavonoid compounds were significantly increased. Genes encoding 2-hydroxyisoflavanone synthase (*CYP93C*), isoflavone 7-O-methyltransferase, isoflavone 4′-O-methyltransferase (*HI4′OMT*), isoflavone/4′-methoxyisoflavone 2′-hydroxylase (*I2′H*), and vestitone reductase (*VR*) were up-regulated, and the synthesis of isoflavones increased significantly. As for ascorbate and aldarate metabolism, L-ascorbate peroxidase (*APX*), which catalyzes the conversion of L-ascorbate to monodehydroascorbate, was down-regulated 3.91-fold, whereas L-ascorbate oxidase (*AO*), which catalyzes the transformation of L-ascorbate into L-dehydroascorbate, was mainly up-regulated (up to 4.75-fold). In addition, glutathione S-transferase (*GST*) was slightly up-regulated. The up-regulation of the proteins isoflavone 4'-O-methyltransferase (HOTHEAD) (3.00-fold) and betaine aldehyde dehydrogenase (*BADH*; 6.01-fold) boosted choline-catalyzed synthesis of betaine. Finally, three pinoresinol/lariciresinol reductase (*PLR*) genes, which catalyze the synthesis of lariciresinol and linseed lignans, were up-regulated. This up-regulation also induced significant enrichment of secondary metabolite biosynthesis pathways.

At the late stage of co-culturing, the up-regulation of genes encoding 4-coumarate-CoA ligase (*4CL*), peroxidase, feruloyl-CoA 6-hydroxylase, and caffeoyl-CoA O-methyltransferase (*CCoAOMT*) continued to promote lignin synthesis. Because two *CHS* and anthocyanidin reductase (*ANR*) coding genes were down-regulated, however, the synthesis of flavonoids decreased. With respect to isoflavone synthesis, only three isoflavone-7-O-methyltransferase genes were up-regulated, which resulted in lower isoflavone accumulation than in the previous period. About the effect of P9 on L-ascorbate metabolism of peanut root, the L-gulonolactone oxidase (*GULO*) coding gene was slightly up-regulated, whereas the *APX* gene was significantly up-regulated (6.28-fold) and the monodehydroascorbate reductase (*MDAR*) gene was significantly down-regulated (−3.62-fold). As a consequence, the scavenging ability of ROS was enhanced. In addition, six genes encoding *GST* were down-regulated (up to 4.85-fold), and the induced expression of polyphenol oxidase and primary-amine oxidase produced more isoquinoline alkaloids.

## RT-qPCR verification

Taking into account the results of GO and KEGG enrichment analyses, we selected 24 genes of peanut co-cultured with strain P9 for 24 h and 72 h, involving cell wall pectin degradation and synthesis, photosynthesis, carbon fixation, the synthesis of ascorbic acid, phenylpropanoid, plant hormone, flavonoids, isoflavones, nitrogen metabolism and other pathways, and verified their expressions by RT-qPCR (Fig 6). A consistent trend was observed in the expression of these 24 genes determined by RT-qPCR and RNA sequencing, indicating that the results of RNA-seq were reliable.

## Discussion

### Strain P9 promotes signal transduction and initiates the defense response of peanut

The MAPK signaling pathway is closely related to plant growth, development, and stress responses [19]. *FLS2*, which binds to bacterial receptor flg22, stimulates ROS production and transmits invasion signals to *WRKY33* to initiate plant defense mechanisms [20]. To have a growth-promoting effect, PGPR must invade plants and activate plant defense functions [21]. *ChiB* is a target gene of *ERF1*, which can induce plant stress responses [22]. After 24 h of co-culturing in our study, genes encoding *FLS2*, *WRKY33*, *ERF1*, and *ChiB* were up-regulated. The initial colonization of strain P9, a foreign microorganism, obviously initiated a rapid plant defense response. Consistent with our findings, Liu et al. found that plant immunity is triggered by the up-regulation of *FLS2*, *WRKY*, *MYB*, and *MYC2* in tobacco roots inoculated with *Paenibacillus polymyxa* YC0136 [23], and the *ERF* transcription factor of *Arabidopsis thaliana* inoculated with *Bacillus amyloliquefaciens* FZB42 is up-regulated [24]. *CNGCs* can transport $Ca^{2+}$, an important second messenger, into the cell and pass it via Rboh and *CaMCML* to induce cell wall thickening and stimulate stomatal closure to resist external environmental pressures [25]. In our study, the up-regulation of *CNGCs* and *CaMCML* during the early stage of co-culturing promoted $Ca^{2+}$ signal transduction and a rapid plant response. After cell injury, Rboh is up-regulated to regulate ROS production and help maintain the dynamic balance of ROS [26]. When we extended the co-culturing time to 72 h, genes encoding *FLS2*, *Rboh*, and *CaMCML* were more strongly up-regulated, whereas the *WRKY33* gene was significantly down-regulated. In addition, the *ChiB* gene was less strongly up-regulated compared with that at the early stage, and the expression of *ERF1* did not differ from that of the control. All of these results indicate that the ethylene-induced plant defense response was weakened and that the regulation of ROS homeostasis induced by strain colonization was stronger.

### Strain P9 stimulates pectin degradation of cell walls to promote cell elongation and facilitates its colonization of peanut

The plant cell wall includes primary and secondary cell walls. The primary cell wall is mainly composed of cellulose, hemicellulose, and pectin. The high pectin polysaccharide content and strong plasticity of plant cells plays an important role in their morphogenesis [27]. *PE*, *PL*, and *PG* can promote the loosening of cell walls, a phenomenon related to cell wall extension and stem elongation [28, 29]. The *PL* gene of Arabidopsis may be involved in cell elongation and lateral root formation [30]. In the present study, the significant up-regulation of the *PE* gene during both time periods and the up-regulation of the *PL* gene at the early stage of co-culturing promoted cell wall elongation and lateral root formation, thereby improving the root development of inoculated peanut (Figs 1 and 2). Because PGPR usually colonize the rhizosphere to exert their growth-promoting effects on plants [31], partial disintegration of cell walls is very important for bacterial adhesion and colonization on root surfaces [32]. The *PL* gene of

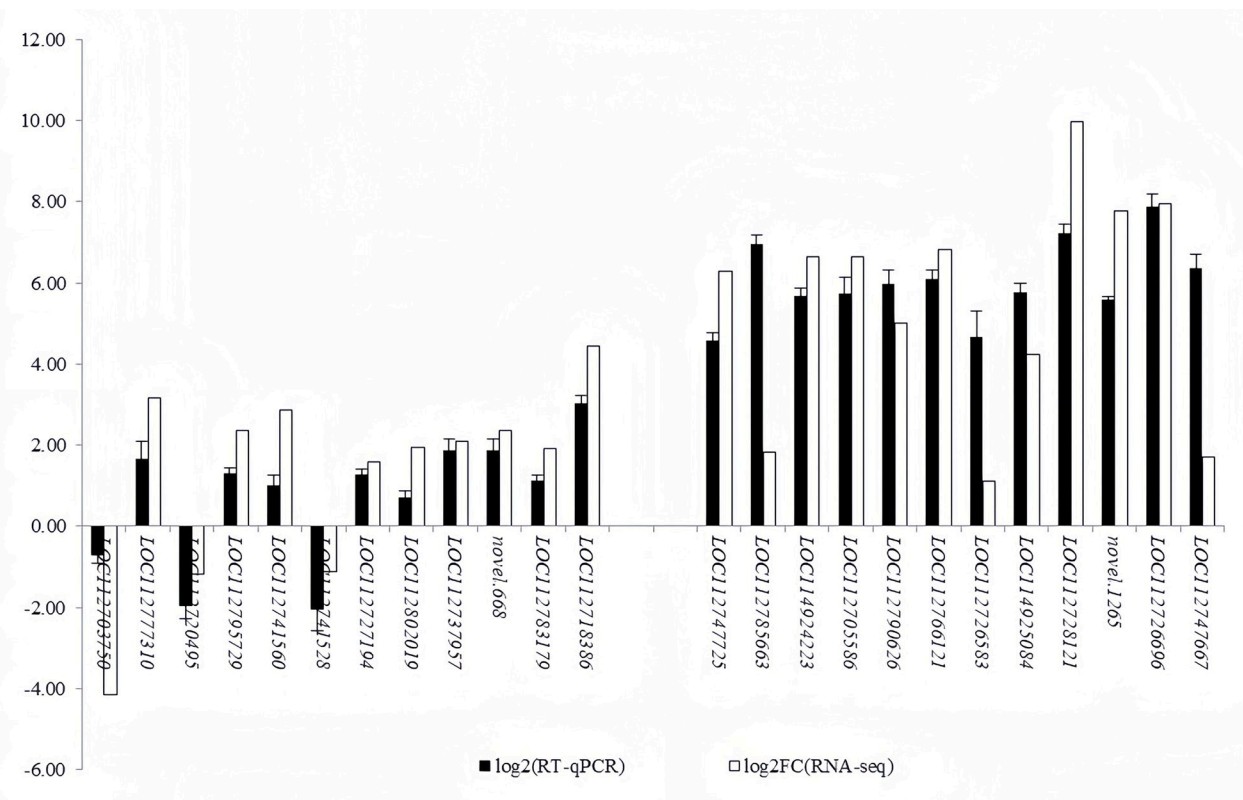

**Fig 6. The differential gene expression verified by RT-qPCR.** The left is the DEGs at 24 h after inoculation; the right is the DEGs at 72 h after inoculation. Among them, *LOC112703750* encodes polygalacturonase, *LOC112777310* encodes trans-cinnamate 4-monooxygenase, *LOC112720495* encodes caffeoyl-CoA O-methyltransferase, *LOC112795729* encodes PetF, *LOC112741560* encodes chalcone synthase-like, *LOC112741528* encodes ent-kaurenoic acid monooxygenase, *LOC112727194* encodes indole-3-pyruvate monooxygenase, *LOC112802019* encodes aminocyclopropanecarboxylate oxidase, *LOC112737957* encodes L-tryptophan decarboxylase, *novel.668* encodes starch synthase, *LOC112783179* encodes ribulose-phosphate 3-epimerase, *LOC112718386* encodes ERF1, *LOC112747725* encodes L-ascorbate peroxidase, *LOC112785663* encodes Nrt, *LOC114924223* and *LOC112705586* encode cytokinin dehydrogenase, *LOC112790626* encodes pectinesterase, *LOC112766121* encodes glucuronokinase, *LOC112726583* encodes L-gulonolactone oxidase, *LOC114925084* encodes 12-oxoophytodienoic acid reductase, *LOC112728121* encodes D-glycerate 3-kinase, *novel.1265* encodes glycerol-3- phosphate acyltransferase, *LOC112726696* encodes CaM4, *LOC112747667* encodes DNA methyltransferase 1.

rapeseed is also up-regulated after the establishment of a symbiotic relationship with the endophytic fungus *Sclerotinia sclerotiorum* strain DT-8 [29]. The PE-coding gene of banana seedlings inoculated with *B. amyloliquefaciens* Bs006 is up-regulated, which promotes pectin degradation and the enhancement of cell adhesion ability, thus facilitating the colonization of banana roots [33]. In our previous study, we found that strain P9 can stably colonize peanut roots and stems [14]. In the present study, we observed that the up-regulated expression of *PE* and *PL* genes in co-cultured peanut increased pectin decomposition, thus laying a foundation for colonization by strain P9.

## Strain P9 affects plant hormone synthesis and transport to promotes peanut growth and development

Strain P9 affects the synthesis of various hormones in peanut plants. In the early stage of co-culturing, we observed that strain P9 induced the up-regulation of *ALDH* and indole-3-pyruvate monooxygenase to increase IAA synthesis and that the up-regulation of the gene encoding *CKX* accelerated the decomposition of cytokinin. Furthermore, the up-regulation of *ACO*, *ACS*, and S-adenosylmethionine synthetase increased ethylene production, and the induced

expression of *LOX*, *OPR3*, acetyl-CoA acyltransferase 1, and hydroperoxide dehydratase promoted the synthesis of jasmonic acid. Finally, the slight down-regulation of ent-kaurenoic acid monooxygenase resulted in a slight decrease in GA synthesis. The development of plant roots is the result of the crosstalk of multiple hormones. IAA plays a leading role in plant root development, with other plant hormones mostly regulated by their coordination or antagonism with IAA [34]. The antagonistic relationship between cytokinin and auxin is well known [35], the balance between these two hormones is a major determinant of cell fate during hormone-induced root organogensis [36], and the overexpression of this *CKX* in rice roots promotes adventitious root formation [37]. In addition, the distribution of auxin will be disturbed by ethylene signal, the effects of ethylene and auxin are synergistic in root elongation and antagonistic in lateral root formation [38]. Ethylene promotes auxin transport to the elongation zone by regulating auxin polar transport proteins. It inhibits the growth of root meristems, promotes the formation of lateral roots, and induces the occurrence of adventitious roots and root hairs [39, 40]. The up-regulation of the *ACS* and *ERF* affects soybean root cortex thickening [41]. GA can regulate stem elongation, and exogenous application of gibberellin has inhibitory effects on root development [42]. Endogenous jasmonic acid can promote the lateral root formation of wild-type *Arabidopsis thaliana* [43]. In our study, increased IAA, ethylene, and jasmonic acid synthesis and decreased cytokinin and GA synthesis obviously promoted peanut root development and the formation of lateral roots and root hairs during the interaction with strain P9. Root lengths and lateral root numbers of seedlings during the early stage of co-culturing were therefore increased compared with those in the control (Figs 1 and 2). Inoculation with *P. fluorescens* significantly increases the concentration of IAA in roots of *Sedum alfredii* and significantly reduces gibberellic acid, trans-zeatin, ethylene, and jasmonic acid concentrations, thereby inducing lateral root formation [44]. In addition, inoculation with *Burkholderia phytofirmans* PsJN induces the up-regulation of IAA, GA, and SA synthesis-related genes in Arabidopsis roots, in turn promoting root hair formation and increased root weights and chlorophyll contents [45]. Moreover, *Paenibacillus polymyxa* YC0136 induces the up-regulation of genes related to IAA, zeatin, and GA synthesis in tobacco roots [23]. Some studies showed that signaling substances secreted by PGPR can affect root development by regulating endogenous hormone pathways of plants [46, 47].

After 72 h of co-culturing, down-regulation of the gene encoding *ACO* had reduced ethylene synthesis, and peanut plants had adapted to colonization by strain P9. Significant up-regulation of *CKX* and *CYP735A* reduced cytokinin levels and promoted root growth [48], and significant up-regulation of *OPR3*, *DOX*, and *ACX* genes increased jasmonic acid synthesis. The down-regulation of gibberellin 2 beta-dioxygenase (*GA2ox*), a key enzyme in the degradation of active gibberellin [49], led to the accumulation of *GA*, and the combined action of the proteins encoded by these genes promoted plant root and stem growth. In particular, the significant down-regulation of the indole-3-pyruvate monooxygenase gene significantly decreased IAA synthesis. Because plants are sensitive to IAA, IAA can act as a signal molecule to interfere with plant hormone synthesis when IAA-producing bacteria interact with plants [50]. IAA secreted by some PGPR strains can also be applied to plants to promote plant growth [51]. In our previous studies, we found that strain P9 has strong IAA-secretion ability [13] and that root exudates of peanut significantly up-regulate the tryptophan metabolic pathway of strain P9 and significantly increase IAA synthesis in this strain [52]. In the present study, we also observed that the genes encoding ABC transporter family proteins *ABCB1* and *ABCB10* were significantly up-regulated in the late stage of co-culturing. The ABCB protein family is mainly located in plant vascular tissue and is closely related to the polar transport of IAA [53]. Auxin signaling and transport play a key role in the root structural changes of *Arabidopsis* stimulated by PGPR strains [54]. We speculate that the IAA secreted by strain P9 and the

enhanced IAA polar transport of peanut seedlings are sufficient to meet and promote plant growth; peanut plants therefore maintain IAA homeostasis by regulating their own IAA synthesis reduction. IAA secreted by *Azospirillum brasilense* has been reported to promote the development of lateral roots and root hairs by regulating IAA homeostasis in plants [55]. In a metabolomic analysis of the interaction co-culture system in our study, we indeed found that the IAA content of the co-culture system increased to varying degrees but was more significant during the early stage of co-culturing. PGPR can regulates the synthesis and homeostasis of plant hormones such as ethylene, auxin, and cytokinin, abscisic acid and gibberellin, which can influence plant growth [56]. So, the combined effects of these hormones significantly promoted seedling root development, and the heights and root characteristics of inoculated plants were significantly improved compared with the findings in the uninoculated controls (Figs 1 and 2).

## Strain P9 enhances peanut metabolism and energy production and promotes plant growth

In the early stage of co-culturing with strain P9, 60 genes involved in circadian rhythm (including *GI*, *CHS*, and *CK2β*), 11 genes participating in the carbon fixation pathway, and 7 genes encoding photosynthetic proteins were significantly up-regulated. Expression of the light-harvesting chlorophyll protein complex LHC subunit was also induced. *GI* is a plant-specific nuclear protein that plays a role in various physiological processes, such as photosynthesis and circadian rhythm [57]. Antenna proteins can capture solar energy and drive photosynthesis [58], while up-regulated Lhcb1 and *Lhcb*3 are directly related to increased chlorophyll content [59]. The *CHS* gene is rhythmically expressed in Arabidopsis seedlings, which makes the regulation of hormones time-dependent [60]. *CK2β* can act on a variety of substrates and affects light signaling, circadian rhythm, and cell cycle control [61]. During the early stage of co-culturing with strain P9, light capture, photosynthesis, carbon fixation, $CO_2$ intake, and circadian rhythm regulation were significantly enhanced in peanut. Moreover, the up-regulation of starch synthase led to an increase in starch synthesis, energy storage, and carbon assimilation products. In addition to light energy, plants can directly produce bio-energy through sugar metabolism. Multiple genes related to starch and sucrose metabolism, glycolysis/gluconeogenesis, amino sugar and ribose metabolism, and pyruvate metabolism were all induced during both co-culturing stages. Genes in pathways related to fatty acid metabolism, acetyl-CoA acyltransferase, acetyl-CoA carboxylase1 *FabH*, long-chain acyl-CoA synthetase, and acyl-CoA thioseterase 1/2/4 were up-regulated, and the synthesis of long-chain fatty acids was increased. Fatty acids are an important source of energy storage in plants and an important component of membrane lipids. Fatty acids can be used as a carbon source by symbiotic mycorrhizal fungi and are considered to be essential for the establishment of symbiotic relationships [62]. Unsaturated fatty acids also maintain the relative fluidity of the cell membrane [63]. Up-regulation of phosphatidate phosphatase LPIN and phospholipid:diacylglycerol acyltransferase increased the production of triglycerides. During the late stage of co-culturing, *GLYK* was significantly up-regulated, and glyceric acid catalyzed the synthesis of large amounts of 3-phosphoglycerate, which was channeled into the glycolysis pathway. Multiple genes involved in sugar metabolism were significantly more highly up-regulated compared with those at the early stage of co-culturing. The production of $CO_2$ and ATP increased significantly, and multiple genes participating in oxidative phosphorylation were also up-regulated, thus indicating that the large amount of ATP obtained by biological oxidation can provide energy for peanut growth and development and various life activities. Moreover, *GPAT* is the first acylesterase used in the synthesis of

phosphatidylglycerol, which significantly up-regulates the synthesis of cell membrane component, namely, 1,2-diacyl-sn-glycerol-3-P. GPAT can also control the formation and storage of membrane lipids in *Arabidopsis thaliana* [64].

During plant growth, roots mainly obtain nitrogen sources by absorbing nitrate and ammonium salts from the soil via nitrate and ammonium transporters. Nitrate and nitrite are transported into cells by nitrate/nitrite transporter (*NRT*), whose activity in roots can affect nitrogen acquisition, plant growth, and seed development [65]. *NR*, the rate-limiting enzyme in nitrate assimilation, is the main compound used by many plants and organisms to obtain nitrogen sources [66]. Overexpression of the *NR* gene can improve the absorption and utilization of nitrogen nutrition in plants [67]. NRT-encoding genes are significantly up-regulated in wheat roots inoculated with *Azospirillum brasilense* inoculated with *Phyllobacterium* strain STM196 [68]. In the present study, the *NR* gene was up-regulated at both co-culturing stages. This gene was significantly up-regulated at the later stage of co-culturing, which indicates that peanut nitrogen uptake and nitrate assimilation were further enhanced and that plants could better obtain sufficient nitrogen sources. AtNRT1.1 is involved in lateral root development and root hair formation in Arabidopsis [69], and *OsNRT2.1* contributes to adventitious root elongation in rice [70]. When the duration of co-culturing with strain P9 was increased, carbon, nitrogen, and lipid metabolism in peanut was therefore accelerated, ATP production was increased, and nitrogen uptake and assimilation were accelerated. Consequently, the promotive effect of P9 on plant growth was more obvious, and root development was better than that of the control group.

Our previous study confirmed that the strain P9 has a good ability to dissolve phosphorus and produce siderophore. Phosphorus is an essential element for the growth and development of plants. In order for plants to use insoluble phosphate in the environment, it must first be dissolved through the action of organic acids [71]. During the early stage of co-culture with strain P9, the PDC and ALDH genes, which catalyse the production of pyruvate and acetic acid, respectively, were up-regulated. Additionally, the LPIN, PDAT, phospholipase C and lysophospholipase II genes, which are involved in glycerophospholipid and glycerolipid metabolism, were also up-regulated. In the later stage, the genes encoding PDC and ADH was up-regulated to a greater extent (7.92-fold and 8.35-fold, respectively), and the genes encoding D-lactate dehydrogenase, malate dehydrogenase and pyruvate decarboxyase, which catalyse the production of lactic acid and malic acid, and the genes encoding choline-phosphate cytase and GPAT, which are involved in the synthesis of phospholipids and phosphoinositides, were significantly up-regulated. Our additional study also confirmed that organic acids are one of the main components of root extracts from peanut seedlings inoculated with the P9 strain [52]. So, it was speculated that the P9 strain not only promoted the formation of organic acids in peanut roots, but also provided more soluble phosphorus for peanuts through its phosphorus solubilsing ability, which obviously promoted plant root development. In addition, iron is closely related to chlorophyll synthesis and photosynthesis, and the regulation of iron homeostasis greatly affects cell signal transduction, which in turn affects root growth [72, 73], but the efficiency of iron utilization by plants is very low. Siderophore secreted by PGPR are an effective strategy for plants to absorb and utilize iron [74]. After inoculation with the P9 strain, the chlorophyll content of the leaves increased, root development was more vigorous, and the expression of photosynthesis-related proteins PetC and PetF was up-regulated. It was speculated that the siderophore secreted by P9 provided sufficient iron for the growth of peanut seedlings, promoting the development of peanut roots and an increase in chlorophyll content. This was consistent with the good plant growth-promoting ability of the P9 strain.

## Strain P9 enhances the defense ability of peanut and maintains ROS homeostasis

The phenylpropanoid pathway is a rich source of plant metabolites; it can not only synthesize lignin, but is also the starting point for the production of flavonoids [75]. This pathway plays an important role in plant resistance to biotic and abiotic stress [76]. The accumulation of lignin, a major component of plant secondary cell walls, helps improve the mechanical strength of cells and is crucial to plant structure and defense [77]. *CYP73A* catalyzes the first oxidation step of the phenylpropanoid pathway in higher plants, whereas *F5H* is the only link in S-type lignin synthesis, and *CCoAOMT* is an important regulatory protein for G-type lignin synthesis [78]. *CAD* catalyzes the last step of lignin biosynthesis before oxidative polymerization of cell walls [79]. *4CL*, a key enzyme involved in the synthesis of lignin and flavonoids, regulates the direction of carbon flow in the downstream branch metabolic pathway of the phenylpropane [80]. In the present study, the up-regulation of *CYP73A*, *F5H*, *CAD*, and *CAGT* at the early stage and that of *4CL*, *CCoAOMT*, and other genes at the late stage all increased lignin synthesis in peanut roots. This increase promoted the formation of secondary structure and vascular bundles, laid a foundation for the development of lateral roots, and enhanced plant mechanical strength. In another study, the up-regulation of *4CL* and *CCoAOMT* genes in tobacco roots after co-culturing with *Paenibacillus polymyxa* YC0136 for 20 h was found to promote cell wall differentiation and thickening and improved stress resistance of tobacco [23].

Flavonoid and isoflavones are secondary metabolites with multiple biological functions. In addition to playing a key role in stress protection and plant defense, these compounds act as signal transduction molecules for the interaction of transcription factors and kinases [81, 82]. During the early stage of co-culturing in our study, 86 differentially expressed genes in the flavonoid and isoflavone biosynthesis pathway were up-regulated compared with those in the control, especially, the first key enzyme in flavonoid biosynthesis, CHS, showed significant induction of up-regulated expression. The up-regulated expression of *CYP93C* and *CYP81E (I2′H)*, members of the cytochrome P450 (*CYP*) superfamily, can protect plants from stress injury [83]. In the early stage of co-culturing in our study, the up-regulation of multiple genes related to flavonoid and betaine synthesis pathways would therefore have significantly improved the defense ability of peanut. In the later stage of co-culturing, the biosynthesis of isoquinoline alkaloids in seedlings was increased. In addition, chitinase plays an important role in plant disease resistance [84]. The Chic-coding gene of peanut co-cultured with strain P9 was up-regulated at both stages, thus indicating that inoculation with this strain improved the disease resistance of peanut.

A complex environment will affect plant growth and development and thus increase ROS content in cells. *APX* and *AO* are important oxidases in the ascorbate metabolic pathway [85]. Most ascorbate present in the cytoplasm binds to *APX* to form monodehydroascorbate and decomposes $H_2O_2$ to remove ROS [86], whereas *AO* mainly oxidizes ascorbate in apoplasts and catalyzes its transformation into L-dehydroascorbate, which is transported to the cytoplasm to maintain the oxidation/reduction state of apoplasts [87]. After co-culturing with strain P9 for 24 h, *AO* was significantly up-regulated, but *APX* was down-regulated, and *APX* and *GULO* genes were up-regulated at 72 h. According to this result, the ability of peanut seedlings to scavenge ROS was significicntly enhanced by prolongation of the strain P9 inoculation time. Additionally, in ~~In~~ the later stage of co-culturing, the significant down-regulation of GST-encoding genes not only maintained the intracellular glutathione (*GSH*) level, but also improved the defense abilities and detoxification of plant cells [88]. In addition, multiple coding genes of peroxidase were significantly up-regulated during the two co-culturing stages. Kandaswamy et al. have reported that *Pseudomonas putida* RRF3 enhances plant defense by

inducing the up-regulation of peroxidase genes in rice roots [89]. Co-culturing with strain P9 obviously enhanced a variety of defense substances and antioxidant enzymes, such as peroxidase, *GST*, *APX*, and *AO* in peanut seedlings, thereby facilitating the rapid removal of excess ROS. The expression of stress-related genes in plants, mediated by PGPR, can improve plant resistance to biotic and abiotic stresses.

## Conclusion

Co-culturing with *Tsukamurella tyrosinosolvens* P9 affected gene expression in peanut seedling roots and promoted plant growth to varying degrees. At 24 and 72 h of co-culturing, the gene expression of peanut seedlings displayed temporal characteristics. After 24 h of co-culturing, the up-regulation of multiple genes in MAPK and $Ca^{2+}$ signal transduction pathways increased ethylene production, which facilitated the rapid response of peanut seedlings to invasion by strain P9. The induced expression of cell wall pectin degradation-related enzymes contributed to colonization by strain P9. By synthesizing more phenylpropane derivatives and flavonoid and isoflavone compounds, the defense ability of peanut plants was enhanced. Plant root development was promoted by increased IAA and jasmonic acid synthesis and the acceleration of cytokinin decomposition. Enhancements in photosynthesis, carbon fixation, circadian rhythm regulation, sugar catabolism, and fatty acid synthesis accelerated seedling growth. The root development of co-cultured seedlings was accordingly superior to that of the control.

After 72 h of co-culturing, ethylene production had decreased as plants adapted to colonization by strain P9. The significant up-regulation of multiple genes in the $Ca^{2+}$ signaling pathway enhanced the regulation of ROS homeostasis in peanut plants. The increased synthesis of isoquinoline alkaloids and the up-regulation of ascorbate and aldarate metabolic pathways improved the ability of plants to scavenge ROS. A decrease in cytokinin synthesis and an increase in jasmonic acid and gibberellin synthesis promoted root and stem growth. Further acceleration of sugar catabolism, up-regulation of oxidative phosphorylation, increased formation of membrane lipids, and enhanced nitrogen metabolism all jointly promoted the uptake and absorption of nutrients by peanut plants. Extension of the co-culturing time therefore significantly increased the growth index of peanut seedlings. As shown by these results, we have clarified the molecular mechanism of peanut response to *Tsukamurella tyrosinosolvens* strain P9, fundamentally analyzed the plant growth-promoting mechanism of P9, and laid a theoretical foundation for the utilization of this strain.

## Supporting information

**S1 Fig.** Plots of differential gene volcanoes in peanut roots inoculated with *Tsukamurella tyrosinosolvens* P9 for 24 h (left) or 72 h (right). Green and red dots indicate down-regulated and up-regulated differentially expressed genes (DEGs), respectively, and blue dots represent genes with no significant expression differences.
(TIF)

**S2 Fig. Principal component analysis (PCA) of peanut.** CK1 and CK2 are control groups at 24 and 72 h, respectively, and T1-P9 and T2-P9 are peanut groups inoculated with P9 for 24 and 72 h, respectively. Each treatment includes three replicates.
(TIF)

**S1 Table. RT-qPCR primer of peanut inoculated with *Tsukamurella tyrosinosolvens* P9 after 24 h and 72 h.**
(XLSX)

**S2 Table. Quality statistics of transcriptome sequencing data in peanut root inoculated with *Tsukamurella tyrosinosolvens* P9 after 24 h and 72 h.**
(XLSX)

**S3 Table. Effect of *Tsukamurella tyrosinosolvens* strains on the expression of genes related to signal transduction in peanut seedlings.**
(XLSX)

**S4 Table. Effect of *Tsukumarella tyrosinosolvens* P9 on the expression of genes related to pectin degradation in peanut root cell walls.**
(XLSX)

**S5 Table. Effect of *Tsukamurella tyrosinosolvens* P9 strain on the expression of hormone synthesis in peanut roots.**
(XLSX)

**S6 Table. Effect of *Tsukamurella tyrosinosolvens* P9 on the expression of lipid metabolism, photosynthesis, carbon and nitrogen metabolism and transporter family in peanut roots.**
(XLSX)

**S7 Table. Effect of *Tsukamurella tyrosinosolvens* P9 strains on the expression of defence- and stress-related genes in peanut roots.**
(XLSX)

## Author Contributions

**Formal analysis:** Xue Bai.

**Funding acquisition:** Lizhen Han.

**Investigation:** Yujie Han.

**Methodology:** Xue Bai.

**Project administration:** Lizhen Han.

**Supervision:** Lizhen Han.

**Writing – original draft:** Xue Bai.

**Writing – review & editing:** Xue Bai, Lizhen Han.

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
