## [Decision Letter · Decision Letter 0]

19 Sep 2023

PONE-D-23-21487Transcriptome alterations of peanut reveal the growth-promoting molecular mechanisms of Tsukamurella tyrosinosolvens P9PLOS ONE

Dear Dr. Lizhen,

Thank you for submitting your manuscript to PLOS ONE. After careful consideration, we feel that it has merit but does not fully meet PLOS ONE’s publication criteria as it currently stands. Therefore, we invite you to submit a revised version of the manuscript that addresses the points raised during the review process.

We look forward to receiving your revised manuscript.

Kind regards,

Anil Kumar Singh, Ph.D.

Academic Editor

PLOS ONE

Journal Requirements:

2. Thank you for submitting the above manuscript to PLOS ONE. During our internal evaluation of the manuscript, we found significant text overlap between your submission and previous work in the [introduction, conclusion, etc.].

Please revise the manuscript to rephrase the duplicated text, cite your sources, and provide details as to how the current manuscript advances on previous work. Please note that further consideration is dependent on the submission of a manuscript that addresses these concerns about the overlap in text with published work.

[If the overlap is with the authors’ own works: Moreover, upon submission, authors must confirm that the manuscript, or any related manuscript, is not currently under consideration or accepted elsewhere. If related work has been submitted to PLOS ONE or elsewhere, authors must include a copy with the submitted article. Reviewers will be asked to comment on the overlap between related submissions (http://journals.plos.org/plosone/s/submission-guidelines#loc-related-manuscripts).]

We will carefully review your manuscript upon resubmission and further consideration of the manuscript is dependent on the text overlap being addressed in full. Please ensure that your revision is thorough as failure to address the concerns to our satisfaction may result in your submission not being considered further.

" This work was supported by the the National Natural Science Foundation of China (32060028). In all of authors, Lizhen Han is the corresponding author, Xue Bai and Yujie Han are Lizhen Han’s graduate students. Lizhen Han designed the experiments, Xue Bai investigated this study and wrote the manuscript, the two students performed the experiments."

"No authors have competing interests."

5. We note that you have referenced (unpublished data) on page 23, which has currently not yet been accepted for publication. Please remove this from your References and amend this to state in the body of your manuscript: (ie “Bewick et al. [Unpublished]”) as detailed online in our guide for authors

Reviewers' comments:

Reviewer's Responses to Questions

**Comments to the Author**

1. Is the manuscript technically sound, and do the data support the conclusions?

Reviewer #1: Partly

Reviewer #2: Partly

2. Has the statistical analysis been performed appropriately and rigorously? 

Reviewer #1: No

Reviewer #2: Yes

3. Have the authors made all data underlying the findings in their manuscript fully available?

Reviewer #1: No

Reviewer #2: No

4. Is the manuscript presented in an intelligible fashion and written in standard English?

Reviewer #1: Yes

Reviewer #2: Yes

5. Review Comments to the Author

Reviewer #1: The authors have presented good information about the changes in the expression levels of different genes which might help to take Tsukamurella tyrosinosolvens strain P9 to the field level. However, major corrections need to be made to improve the manuscript quality.

Differential expression alone cannot reveal any molecular interaction or mechanism. The title must be changed according to the study conducted. This study only explains the changes in the gene expression during the P9 colocalization of peanut roots. No molecular mechanism has been revealed here.

Authors are requested to provide the Bioproject ID or GEO ID for the submitted RNAseq data in the data availability statement. A supplementary file containing the list of all the DEGs should be included.

Why the authors have used LOC IDs when the full genome sequence of peanuts is available in the database (https://www.peanutbase.org/) is unclear.

Line 88- 105: The catalog number of the products can be mentioned here. Mention of the software used for read cleaning and quality filtering, pathway enrichment studies, and GO is required.

Line 109-119: Tool used for primer designing needs to be mentioned. As qPCR is a very generic experiment, details of the protocol can be omitted. Mention the citation for 2-∆∆Ct method.

Line 97: Mention the table number where the data has been provided.

Line 137-138: Selection of two time points needs better justification. At least, the initial phenotyping experiments could have been done in three or more time points (24hrs, 48hrs, 72hrs) to determine the stage at which significant differences are observed. Highly significant differences in the root phenotypic characters may indicate an induction of these changes at an earlier time point.

Line 167-175: Please indicate the number of common DEGs due to the addition of P9 strain. Are there any unique genes that are differentially expressed during initial (24hr) and final (72hrs) treatment? Use of a vein diagram rather than a volcano plot might be helpful in this case.

Line 220-222, 243-245, 333: Sentence reconstruction required for correct understanding

Line 214, 252: citation required

Line 303: The authors commented on enhancement in photosynthesis rate. However, no physiological data has been provided. Moreover, the enhancement in chlorophyll content was not significant. Please comment on this.

Line 380-383: This part has already been mentioned in Materials and Methods section. The result section should have the interpretation of the data.

Fig S1: Statistics required in the bar graph.

Suggestion: As a lot of hormonal changes have been observed at the transcript level, a critical analysis of cell division and elongation genes can give better insight.

Discussion section could have been written in a better pattern. Mentioning a lot of gene regulations in the first part and connecting them to relevant papers in the later part will confuse the readers. Re-orienting sentences in a connected manner will improve the content.

Expanded forms of the abbreviations used for the first time must be included for easy understanding though they are present in the supplementary file.

Claims leading to improved accumulation or resistance should be toned down as no evidence for the same has been provided in this study. Mere upregulation or downregulation of genes cannot suggest the final activity in the mature stage of the plant.

Suggestion: Volcano plots or PCA analysis figures can be provided in the supplementary files. A heatmap or pictorial representation of the analyzed genes should be presented instead which can provide consolidated information to the readers. A graphical abstract about the important findings of this colocalization study can help the readers too.

This manuscript follows the same pattern as the ones published earlier on related topics. This creates redundancy, which must be avoided. The content and pattern of this manuscript are largely inspired by these two papers:

Jiang, B., Long, C., Xu, Y., & Han, L. (2023). Molecular mechanism of Tsukamurella tyrosinosolvens strain P9 in response to root exudates of peanut. Archives Of Microbiology, 205(1). doi: 10.1007/s00203-022-03387-7

Han, L., Zhang, H., Bai, X., & Jiang, B. (2023). The peanut root exudate increases the transport and metabolism of nutrients and enhances the plant growth-promoting effects of burkholderia pyrrocinia strain P10. BMC Microbiology, 23(1). doi: 10.1186/s12866-023-02818-9

Reviewer #2: The study titled "Transcriptome Alterations of Peanut Revealing the Growth-Promoting Molecular Mechanisms of Tsukamurella tyrosinosolvens P9," conducted by Lizhen Han and and co-workers established a a co-culture system between Tsukamurella tyrosinosolvens strain-P9 and peanut plants. They performed a transcriptome analysis of peanut roots interacting with strain P9 at two different time points. The primary aim was to investigate how peanut seedlings respond at the molecular level to the influence of strain P9 and to provide direct evidence supporting an in-depth analysis of the growth-promoting mechanisms employed by strain P9 on peanuts.

Their previous study reported Tsukamurella tyrosinosolvens P9 exhibits strong phosphate solubilization, indole-3-acetic acid (IAA) secretion, and siderophore production. Which transcripts are directly related to phosphate solubilization and siderophore production? The growth-promoting molecular mechanism of strain P9 is not clear in the present investigation, which which should be addressed in future revisions.

Why the authors chose the genome of Arachis hypogaea (assembly arahy.Tifrunner.gnm1.KYV3; GenBank accession number GCA_003086295.2) as a reference guided identification of transcriptome from the RNA-seq data. Silihong, an early maturing, local peanut variety with a dark red seed coat was used as the plant material in the present investigation. Based on the classification of cultivated peanuts, the “Silihong” variety belongs to which group (Spanish bunch type or Verginia runner type), that may be mentioned. Given the published and publicly available genome sequences of these subgroups of the Arachis species [Zhuang, Weijian, et al. Nature genetics 51, no. 5 (2019): 865-876; Bertioli, David J. et al. Nature genetics 51, no. 5 (2019): 877-884; Chen, Xiaoping, Molecular plant 12, no. 7 (2019): 920-934] selecting the appropriate reference genome is crucial for accurate transcriptome analysis.

The authors should explicitly define the rationale for selecting only 12 genes for qRT-PCR validation except for the GO (Gene Ontology) and KEGG (Kyoto Encyclopedia of Genes and Genomes) analyses. Moreover, qRT-PCR validation is an important aspect of any transcriptomics study. The data needs to be incorporated in the main text instead of supplementary data. If possible the raw qRT PCR experimental data may be shown in the supplementary file. Additionally, the authors should clarify whether there are any known connections between these 12 genes in the growth-promoting molecular mechanisms of the Tsukamurella tyrosinosolvens P9 strain. A comprehensive analysis of the transcriptome data except for the discussion of log2 fold change data is lacking in the present investigation.

While the authors claim that the DEG (differentially expressed gene) and KEGG (Kyoto Encyclopedia of Genes and Genomes) enrichment analyses revealed spatiotemporal differences in gene expression in peanut roots in response to strain P9, they have not provided experimental evidence to support this claim. Furthermore, the qRT-PCR validation was conducted only on root transcripts, and the study lacks validation in other tissues to confirm the spatiotemporal distribution of the identified transcripts from the RNA-seq data.

The raw RNA-seq data need to be submitted to the data repository and should also be available.

Lastly, there are some minor corrections needed in the manuscript, such as using "indices" instead of "indexes" in line numbers 141 and 162, italicizing the scientific name "Tsukamurella tyrosinosolvens" consistently throughout the manuscript, and italicizing gene names. Additionally, "Ca2+" should be written correctly in the whole manuscript.

6. PLOS authors have the option to publish the peer review history of their article (what does this mean?). If published, this will include your full peer review and any attached files.

Reviewer #1: **Yes: **Rajarshi Sanyal

Reviewer #2: No

---

## [Author Response · Author response to Decision Letter 0]

7 Nov 2023

Dear Editor,

Please find attached a revised version of our manuscript “Transcriptional alterations of peanut root during interaction with growth-promoting Tsukamurella tyrosinosolvens strain P9”.

Thank you for your kind letter regarding our manuscript, and the referees’ remarks. Your comments and those of the reviewers were highly insightful and enabled us to greatly improve the quality of our manuscript. We have revised and modified the MS in accordance with the reviewers’ comments, and revisions in the text are shown using yellow highlight for additions, and strikethrough font for deletions. And we also responded point-by-point to each reviewer comments as listed below. Accordingly, we changed the serial number of the Reference because of some reference’s deletion and addition.

And about this manuscript financial disclosure:

This work was supported by the the National Natural Science Foundation of China (32060028). The funders had no role in study design, data collection and analysis, decision to publish, or preparation of the manuscript. In all of authors, Lizhen Han is the corresponding author, Xue Bai and Yujie Han are Lizhen Han’s graduate students. Lizhen Han designed the experiments, Xue Bai investigated this study and wrote the manuscript, the two students perfomed the experiments.

We hope that the revisions in the manuscript and our accompanying responses will be sufficient to make our manuscript suitable for publication in PLOS ONE. We shall look forward to hearing from you at your earliest convenience.

Yours sincerely,

First author, Xue Bai; 

Corresponding author, Lizhen Han.

Corresponding author E-mail: lzhan1@gzu.edu.cn; hanlizhen11@163.com

Responses to the comments of Reviewer 1

1.Differential expression alone cannot reveal any molecular interaction or mechanism. The title must be changed according to the study conducted. This study only explains the changes in the gene expression during the P9 colocalization of peanut roots. No molecular mechanism has been revealed here.

Response:

Thanks for your suggestion. The title has been modified to “”Transcriptional alterations of peanut root during interaction with growth-promoting Tsukamurella tyrosinosolvens strain P9”. Accordingly, we changed “transcriptional alterations of peanut root during interaction with P9” instead of “the molecular mechanism of peanut in response to P9” in Line28-29 in Abstract. 

2. ID for the submitted RNAseq data in the data availability statement. A supplementary file containing the list of all the DEGs should be included.

Response:

Thanks, the RNA-seq raw data has been uploaded to the NCBI database, the accession number is PRJNA991079 (https://www.ncbi.nlm.nih.gov/sra/PRJNA991079); and this section has been added to Line110-112 of the manuscript.

3.Why the authors have used LOC IDs when the full genome sequence of peanuts is available in the database (https://www.peanutbase.org/) is unclear.

Response:

I’m so sorry about an imprecise expression. This reference genome has the full genome sequence, we miswrote the URL of the peanut’s genome. So, we recorrected it in Line103-104, and added one cited reference after this section.

Tang K, Li L, Zhang B, et al. (2023). Gene co-expression network analysis identifies hub genes associated with different tolerance under calcium deficiency in two peanut cultivars. BMC Genomics, 24(1), 421. 

4.Line 88-105: The catalog number of the products can be mentioned here. Mention of the software used for read cleaning and quality filtering, pathway enrichment studies, and GO is required.

Response: 

Thanks for your suggestion. We added the software used for read cleaning and quality filtering, GO and KEGG analysis in Line100, Line105, and Line111-112. And we added accession number of our RNA-Seq raw data in NCBI database in Line112-113.

5.Line 109-119: Tool used for primer designing needs to be mentioned. As qPCR is a very generic experiment, details of the protocol can be omitted. Mention the citation for 2-∆∆Ct method

Response: 

Primer-BLAST was used for primer design, which was mentioned in Line117-118. The experimental procedure of qPCR has been deleted in Line 118-125. And we added the citation for 2-∆∆Ct method in Line127. 

Livak KJ, & Schmittgen TD. (2001). Analysis of relative gene expression data using real-time quantitative PCR and the 2(-Delta Delta C(T)) Method. Methods (San Diego, Calif.), 25(4), 402-408. 

. 

6.Line 97: Mention the table number where the data has been provided. 

Response:

Because there is only the “Materials and Methods”, so we did not add (S2 Tab) in it. However, this message was seen in the corresponding “Results” in Line153. 

7.Line 137-138: Selection of two time points needs better justification. At least, the initial phenotyping experiments could have been done in three or more time points (24hrs, 48hrs, 72hrs) to determine the stage at which significant differences are observed. Highly significant differences in the root phenotypic characters may indicate an induction of these changes at an earlier time point.

Response:

Thanks for your opinion, of course this will provide additional and meaningful data to support dynamic changes at the transcriptome level after inoculation. In fact, we have done a pre-experiment of dynamic observation and measurement, we found that after inoculation with P9 strain 48h, there was no significant difference in the growth and physiological indexes of peanut compared with the uninoculated seedlings (seen in below figure), while there was obvious difference on the peanut’s of 72 h inoculation compared with the uninoculated control. Therefore, in order to better correspond to the changes of physiological and morphological indices, we chose these two time points (24h, 72h) to further analyze the changes at the transcriptomic level. 

24h inoculation 48h inoculation 72h inoculation

Fig Dynamic changes of peanut inoculated with P9 strain

(left is CK, right is seedlings inoculated with P9)

8. Line 167-175: Please indicate the number of common DEGs due to the addition of P9 strain. Are there any unique genes that are differentially expressed during initial (24hr) and final (72hrs) treatment? Use of a vein diagram rather than a volcano plot might be helpful in this case.

Response:

Thanks for your suggestion. We added two venn diagrams (Fig 3.) in Line200-203 and related description in Line182-190. Accordingly, volcanoes plots was put in S Figure. 

9. Line 220-222, 243-245, 333: Sentence reconstruction required for correct understanding; Line 214, 252: citation required

Response:

We reconstructed these three sentences, including in Line248-251 about “Ca2+” and related section, in Line269-274 about “CKX gene” and related section, in Line363-364 about “ATPase, NdhK and Ndufb7” and related section. In addition, we added the citation in Line242 and Line282, and accordingly, these two citation were added in the “References”.

 Xu J, Zhang S. (2015) Mitogen-activated protein kinase cascades in signaling plant growth and development. Trends Plant Sci. 2015;20(1):56-64. 

Watanabe M, Chiba Y, & Hirai MY. (2021) Metabolism and Regulatory Functions of O-Acetylserine, S-Adenosylmethionine, Homocysteine, and Serine in Plant Development and Environmental Responses. Frontiers in plant science, 12, 643403. 

10.Line 303: The authors commented on enhancement in photosynthesis rate. However, no physiological data has been provided. Moreover, the enhancement in chlorophyll content was not significant. Please comment on this.

Response:

Thanks for your suggestion. Because it was an imprecise expression about “photosynthesis rate”, we deleted “increased the rate of photosynthesis” and changed it with “photosynthesis” in Line336. Regarding photosynthesis and chlorophyll content, it is well known that photosynthesis consists of a light reaction and a dark reaction, the light reaction is the absorption and transfer of light energy and conversion (primary reaction), then the production of ATP by electron transfer and photophosphorylation, and finally the dark reaction by the fixation of CO2. Chloroplast membrane contains photosynthetic pigments and electron transport chain components, which perform the conversion of light energy to chemical energy. In this study, the photosynthesis pathway and antenna protein pathway were up-regulated, including photosystem II (PSII) oxygen-evolving enhancer protein (PsbO，PsbP，PsbQ)，photosystem I (PSI) subunit II (PsaD)、cytochrome b6-f complex iron-sulfur subunit (PetC), photosynthethic electron transport ferredoxin (PetF)、light-harvesting complex II (LHCII) chlorophyll a/b binding protein (Lhcb1, Lhcb3), their up-regulated fold is about 1.07-2.70. The results show that the absorption and conversion of light energy and electron transfer are enhanced, and the light reaction is accelerated. Accordingly, chlorophyll content was increased slightly, which is consistent with the change of gene expression on the photosynthesis and antenna protein. We adjusted this section in Line330-341 to make it clearer. In addition, It was shown that plants can regulate the expression of key enzymes involved in photosynthesis through circadian rhythms and ultimately promote plant growth (Li et al. 2020), these consisted with significant enrichment in the circadian rhythms in our study.

Li Z, Zhu A, Song Q, et al. (2020). Temporal regulation of the metabolome and proteome in photosynthetic and photorespiratory pathways contributes to maize hetorosis. Plant Cell, 32(12): 3706-3722.

11.Line 380-383: This part has already been mentioned in Materials and Methods section. The result section should have the interpretation of the data.

Response:

Based on two reviewers’ opinion, we added Figure of RT-qPCR (Fig 7.) and some interpretation of these data in Line417-438.

12. Fig S1: Statistics required in the bar graph.

Response:

In the RNA-Seq data, in the same treated group, the analysis of DEG is based on the normalization of original readcount of three samples, and then the determination of P-value and Padj values by statistical model, then log2FoldChange was calculated by the ratio of mean gene expression levels between the treatment group and the control group. So, the log2FoldChange from RNA-Seq was the only one value, there were also related references such as Liu et al. (2021) and Liu et al. (2023). Of course, our RT-qPCR value has the bar graph. 

Liu H, Li Y, Ge K, et al. (2021). Interactional mechanisms of Paenibacillus polymyxa SC2 and pepper (Capsicum annuum L.) suggested by transcriptomics. BMC Microbiology, 21: 70.

Liu H, Wang J, Sun H, et al. (2020). Transcriptome profiles reveal the growth-promoting mechanisms of Paenibacillus polymyxa YC0136 on tobacco (Nicotiana tabacum L.). Frontiers in Microbiology, 11: 584174. 

13. Suggestion: As a lot of hormonal changes have been observed at the transcript level, a critical analysis of cell division and elongation genes can give better insight.

Response:

Thanks for your suggestion. We modified, deleted and added some sentences about this section, including Line486, Line495-505, Line521-523, and Line549-550; and accordingly, these citations were added in the References.

Mao C, He J, Liu L, Deng Q, Yao X, Liu C, et al. OsNAC2 integrates auxin and cytokinin pathways to modulate rice root development. Plant Biotech. 2020;18(2):429–442. 

Lakehal A, Bellini C. Control of advenitious root formation: insights into synergistic and antagonistic hormonal interactions. Physiol Plantarum. 2019;165:90-100. 

Qin H, Huang R. Auxin Controlled by Ethylene Steers Root Development. Int J Mol Sci. 2018;19(11): 3656. 

Swarup R, Perry P, Hagenbeek D, Van Der Straeten D, Beemster GT, Sandberg G, et al. Ethylene upregulates auxin biosynthesis in Arabidopsis seedlings to enhance inhibition of root cell elongation. The Plant Cell. 2007;19(7):2186–2196. 

Moreno-Risueno MA, Van Norman JM, Moreno A, Zhang J, Ahnert SE, Benfey PN. Oscillating gene expression determines competence for periodic Arabidopsis root branching. Science. 2010;329(5997):1306–1311. 

Du Y, Scheres B. Plethora transcription factors orchestrate de novo organ patterning during Arabidopsis lateral root outgrowth. PNAS. 2017;114(44):11709–11714. 

Zamioudis C, Mastranesti P, Dhonukshe P, Blilou I, Pieterse CM. Unraveling root developmental programs initiated by beneficial Pseudomonas spp. bacteria. Plant Physio, 2013;162(1):304–318. 

Sun X, Wang N, Li P, Jiang Z, Liu X, Wang M, et al. Endophytic fungus Falciphora oryzae promotes lateral root growth by producing indole derivatives after sensing plant signals. Plant, Cell & Environ. 2020;43(2):358–373. 

14.Discussion section could have been written in a better pattern. Mentioning a lot of gene regulations in the first part and connecting them to relevant papers in the later part will confuse the readers. Re-orienting sentences in a connected manner will improve the content.

Response:

Thank you for your suggestion. In fact, as for the writing mode of this part, we have considered writing results and discussions together, we also have considered these results and discussions in our study combining with cited related reference; However, there were so many differentially expressed genes and multiple metabolic pathways involved in two inoculation period (24 h and 72 h, P9 inoculation and no-inoculation, respectively), it's more likely to be confused. So, finally we wrote the results and discussion separately, which is clearer and easier to understand.

15.Expanded forms of the abbreviations used for the first time must be included for easy understanding though they are present in the supplementary file.

Response:

We have checked carefully and the abbreviations had been added their expanded forms.

16.Claims leading to improved accumulation or resistance should be toned down as no evidence for the same has been provided in this study. Mere upregulation or downregulation of genes cannot suggest the final activity in the mature stage of the plant.

Response:

Thanks for your suggestion. We deleted and changed some describtion about “accumulation of anti-stress substances or bioactive substances” in Line376-381, Line385-393, Line622-637. 

17.Suggestion: Volcano plots or PCA analysis figures can be provided in the supplementary files. A heatmap or pictorial representation of the analyzed genes should be presented instead which can provide consolidated information to the readers. A graphical abstract about the important findings of this colocalization study can help the readers too.

Response:

Thanks, volcanoes plots and PCA analysis figures were put in Supplementary “S Figure”, these related description was deleted, too. And we added heatmap (Fig 4.) in Line205-207 and some sentences in Line192-195.

18.This manuscript follows the same pattern as the ones published earlier on related topics. This creates redundancy, which must be avoided. The content and pattern of this manuscript are largely inspired by these two papers:

Jiang, B., Long, C., Xu, Y., & Han, L. (2023). Molecular mechanism of Tsukamurella tyrosinosolvens strain P9 in response to root exudates of peanut. Archives Of Microbiology, 205(1). doi: 10.1007/s00203-022-03387-7

Han, L., Zhang, H., Bai, X., & Jiang, B. (2023). The peanut root exudate increases the transport and metabolism of nutrients and enhances the plant growth-promoting effects of Burkholderia pyrrocinia strain P10. BMC Microbiology, 23(1). doi: 10.1186/s12866-023-02818-9

Response:

Thanks for your carfully review. Both this manuscript and these two references above were completed by our laboratory. Transcriptome sequencing is a common technique at present. Tsukamurella tyrosinosolvens P9 and Burkholderia pyrrocinia P10 were excellent PGPR strains isolated and screened by our lab, which could promote obviously the growth of peanut seedlings and also had complex interaction with peanut. Therefore, we have conducted in-depth and comprehensive research on these two strains. In these two references, it was studied that the effects of peanut root exudates and its components on the transcriptome of these two strains. In this manuscript, by constructing peanut-strain interaction co-culture system, we want to analyze the effects of P9 strain on the gene expression of peanut seedlings, aim to explore the promoting-growth mechanism of P9 on peanut. Although transcriptome analysis was adopted in both systems, but RNA-Seq was conducted on strains and peanuts, respectively, and the former was analyzed under in vitro conditions, while the latter was analyzed in the interaction system, so I thought there were obvious different and without redundancy.

Responses to the comments of Reviewer 2

1.Their previous study reported Tsukamurella tyrosinosolvens P9 exhibits strong phosphate solubilization, indole-3-acetic acid (IAA) secretion, and siderophore production. Which transcripts are directly related to phosphate solubilization and siderophore production? The growth-promoting molecular mechanism of strain P9 is not clear in the present investigation, which which should be addressed in future revisions.

Response:

It has been confirmed that the strain P9 has good ability to dissolve phosphorus and produce siderophore in our previous study. Phosphorus is an essential element for the growth and development of plant. After P9 inoculation, the genes related to phospholipid and inositol phosphorus synthesis in peanut roots were up-regulated, it showed that the available phosphorus of peanut increased. We speculate that it may be related to dissolving phosphorus capabilty of P9 strain, which could provide more phosphorus to roots of peanut seedlings. In addition, iron is closely related to chlorophyll synthesis and photosynthesis, the regulation of iron homeostasis greatly affects cell signal transduction and further affects the growth of root (Tsai & Schmidt 2017; Kobayashi et al. 2019), however, iron utilization efficiency of plants is very low. Siderophore secreted by PGPR is an effective strategy for plants to absorb and utilize iron (Liang 2022). After inoculation with P9 strain, the chlorophyll content of leaves increased, the development of roots more vigorous, and the expression of photosynthesis-related proteins was up-regulated. So, we speculate that the siderophore secreted by P9 provided sufficient iron for the growth of seedlings, and promoted the root development and nutrient absorption of peanut. And in fact the growth-promoting molecular mechanism of strain P9 is not entirely clear, we need to do more research in future revisions, and thanks for your suggestion.

Kobayashi T, Nozoye T, Nishizawa NK. (2019). Iron transport and its regulation in plants. Free Radical Biology & Medicine, 133: 11-20. 

Liang G. (2022). Iron uptake, signaling, and sensing in plants. Plant Communications, 3(5): 100349. 

Tsai HH, & Schmidt W. (2017). One way, or another? Iron uptake in plants. The New phytologist, 214(2): 500-505. 

2.Why the authors chose the genome of Arachis hypogaea (assembly arahy.Tifrunner.gnm1.KYV3; GenBank accession number GCA_003086295.2) as a reference guided identification of transcriptome from the RNA-seq data. Silihong, an early maturing, local peanut variety with a dark red seed coat was used as the plant material in the present investigation. Based on the classification of cultivated peanuts, the “Silihong” variety belongs to which group (Spanish bunch type or Verginia runner type), that may be mentioned. Given the published and publicly available genome sequences of these subgroups of the Arachis species [Zhuang, Weijian, et al. Nature genetics 51, no. 5 (2019): 865-876; Bertioli, David J. et al. Nature genetics 51, no. 5 (2019): 877-884; Chen, Xiaoping, Molecular plant 12, no. 7 (2019): 920-934] selecting the appropriate reference genome is crucial for accurate transcriptome analysis.

Response:

Thanks for your kindly suggestion. So far, the peanut genome has nine in GenBank and arahy. Tifrunner. Gnm1. KYV3 comes from the cultivar Tifrunner, submitted by the international peanut genome initiative on May 30, 2018. It was reported that Tifrunner is a runner-type peanut (Bertioli et al. 2019). Wang et al. (2016) reported that Silihong belongs to the genus Arachis hypocotyl subspecies of family Fabaceae, in which lacks flowers on the main axis, Verginia Runner type belongs to hypocotyl subgenus. In this study, the percentage of genome sequences of Silihong matching the reference genome of Tifrunner was higher than 94.34%. Based on genome charateristics and morphology of plant and seed (Zhuang et al. 2019), Silihong belonged to Verginia runner type. And this type was added in Line65.

Bertioli DJ, Jenkins J, Clevenger J, et al. (2019). The genome sequence of segmental allotetraploid peanut Arachis hypogaea. Nature Genetics, 51(5): 877-884.

Wang H, Khera P, Huang B, et al. (2016). Analysis of genetic diversity and population structure of peanut cultivars and breeding lines from China, India and the US using simple sequence repeat markers. Journal of Integrative Plant Biology, 58(5):452-465.

Zhuang W, Chen H, Yang M, et al. (2019). The genome of cultivated peanut provide insight into legume karyotypes, polyploid evolution and crop domestication. Nature genetics 51(5): 865-876.

3.The authors should explicitly define the rationale for selecting only 12 genes for qRT-PCR validation except for the GO (Gene Ontology) and KEGG (Kyoto Encyclopedia of Genes and Genomes) analyses. Moreover, qRT-PCR validation is an important aspect of any transcriptomics study. The data needs to be incorporated in the main text instead of supplementary data. If possible the raw qRT PCR experimental data may be shown in the supplementary file. Additionally, the authors should clarify whether there are any known connections between these 12 genes in the growth-promoting molecular mechanisms of the Tsukamurella tyrosinosolvens P9 strain. A comprehensive analysis of the transcriptome data except for the discussion of log2 fold change data is lacking in the present investigation.

Response:

Based on the expression of DEGs and effects on the KEGG pathways, we selected 24 genes, involving cell wall pectin degradation and synthesis, photosynthesis, carbon fixation, the synthesis of ascorbic acid, phenylpropanoid, plant hormone, flavonids and isoflavones, nitrogen metabolism and other pathways, and verified their expressions by RT-qPCR. These section was added in Line417-423; and the figure and illustration of qRT-PCR (Fig 7.) has been added in Line424-438. The specific gene names and involved metabolic pathways has been in Supplementary Table 2, and the original data of RT-qPCR in Supplementary “Raw data and analysis of Supp Fig. 1” of originial manuscript. RT-qPCR is mainly used for the validation of transcriptome data, we did not consider deeply between these genes in the growth-promoting molecular mechanisms of the Tsukamurella tyrosinosolvens P9 strain. However, this is a good suggestion and idea to further study growth-promoting molecular mechanisms of P9 strain. In Conclusion, we conduct a general analysis of the transcriptome and found preliminary some interesting and possible mechanism, however, the molecular mechanisms on the effects of peanut by P9 strain was needed in future study.

4.While the authors claim that the DEG (differentially expressed gene) and KEGG (Kyoto Encyclopedia of Genes and Genomes) enrichment analyses revealed spatio temporal differences in gene expression in peanut roots in response to strain P9, they have not provided experimental evidence to support this claim. Furthermore, the qRT-PCR validation was conducted only on root transcripts, and the study lacks validation in other tissues to confirm the spatiotemporal distribution of the identified transcripts from the RNA-seq data.

Response:

Thanks for your suggestion. I’m so sorry about an imprecise expression, We changed “temporal differences” instead of “spatial and temporal differences” in Line28 in Abstract, and changed “Temporal difference” instead of “Spatial and temporal difference” in Keywords, and deleted “-temporal” in Line192. The reasons that we selected the peanut root for transcriptome analysis was as follows: In this co-culture system, the peanut root was dipped into the bacterial suspension, which is more likely to interact with P9 strain and have a more obvious impact on the gene expression of the root in a short period of time. In fact, from the morphological observation and growth index measurement, we could found that the root development after inoculated with P9 strain was more stronger than those of CK, and their root length, root weight and lateral root number were significantly different from those of CK (P<0.01). Subsequent studies on the influence of P9 strain on gene expression in other tissues of peanut will obtain more information on spatial differences. 

5.The raw RNA-seq data need to be submitted to the data repository and should also be available.

Response:

The RNA-seq raw data has been uploaded to the NCBI database, the accession number is PRJNA991079 (https://www.ncbi.nlm.nih.gov/sra/PRJNA991079). This section has been added to Line 112-113. 

6.Lastly, there are some minor corrections needed in the manuscript, such as using "indices" instead of "indexes" in line numbers 141 and 162, italicizing the scientific name "Tsukamurella tyrosinosolvens" consistently throughout the manuscript, and italicizing gene names. Additionally, "Ca2+" should be written correctly in the whole manuscript.

Response:

 We changed “indices” instead of “indexes” in Line149. In addition, scientific name of this species was put in italics, Ca2+ was wrote in superscript, too.

---

## [Decision Letter · Decision Letter 1]

29 Nov 2023

PONE-D-23-21487R1Transcriptional alterations of peanut root during interaction with growth-promoting  Tsukamurella tyrosinosolvens  strain P9PLOS ONE

Dear Dr. Lizhen,

Thank you for submitting your manuscript to PLOS ONE. After careful consideration, we feel that it has merit but does not fully meet PLOS ONE’s publication criteria as it currently stands. Therefore, we invite you to submit a revised version of the manuscript that addresses the points raised during the review process.

Both the reviewers still have some minor suggestions for the improvement of the manuscript. Please submit your revised manuscript by Jan 13 2024 11:59PM. If you will need more time than this to complete your revisions, please reply to this message or contact the journal office at plosone@plos.org. Please include the following items when submitting your revised manuscript:A rebuttal letter that responds to each point raised by the academic editor and reviewer(s). You should upload this letter as a separate file labeled 'Response to Reviewers'.A marked-up copy of your manuscript that highlights changes made to the original version. You should upload this as a separate file labeled 'Revised Manuscript with Track Changes'.An unmarked version of your revised paper without tracked changes. You should upload this as a separate file labeled 'Manuscript'.If applicable, we recommend that you deposit your laboratory protocols in protocols.io to enhance the reproducibility of your results. Protocols.io assigns your protocol its own identifier (DOI) so that it can be cited independently in the future. For instructions see: https://journals.plos.org/plosone/s/submission-guidelines#loc-laboratory-protocols. Additionally, PLOS ONE offers an option for publishing peer-reviewed Lab Protocol articles, which describe protocols hosted on protocols.io. Read more information on sharing protocols at https://plos.org/protocols?utm_medium=editorial-email&utm_source=authorletters&utm_campaign=protocols.

We look forward to receiving your revised manuscript.

Kind regards,

Anil Kumar Singh, Ph.D.

Academic Editor

PLOS ONE

Journal Requirements:

Reviewers' comments:

Reviewer's Responses to Questions

**Comments to the Author**

1. If the authors have adequately addressed your comments raised in a previous round of review and you feel that this manuscript is now acceptable for publication, you may indicate that here to bypass the “Comments to the Author” section, enter your conflict of interest statement in the “Confidential to Editor” section, and submit your "Accept" recommendation.

Reviewer #1: (No Response)

Reviewer #2: All comments have been addressed

2. Is the manuscript technically sound, and do the data support the conclusions?

Reviewer #1: Yes

Reviewer #2: Partly

3. Has the statistical analysis been performed appropriately and rigorously? 

Reviewer #1: Yes

Reviewer #2: Yes

4. Have the authors made all data underlying the findings in their manuscript fully available?

Reviewer #1: No

Reviewer #2: Yes

5. Is the manuscript presented in an intelligible fashion and written in standard English?

Reviewer #1: Yes

Reviewer #2: Yes

6. Review Comments to the Author

Reviewer #1: I appreciate the effort the authors have put into incorporating the suggested changes. It's evident that the authors have taken the feedback seriously, and the manuscript is progressing well. However, I would like to draw your attention to sentence construction and grammatical errors.

1. There are still some grammatical errors and spelling mistakes throughout the manuscript (for example: Line 58, 182, 185, 192, 188, 205, 250, 332, 333, 364, 367, 377, 420, 466, 499, 505, 549, 558, 561, 634, 651, 656, 658, 665). The use of the phrase, “in regard to” has been used excessively, making the story monotonous. I suggest using a grammar checking software to resolve this.

2. Line 65: Please correct the spelling of peanut type. It should be “Virginia runner”

3. The link given in data availability statement is not working as of 27.11.23.

Overall, I believe the manuscript is moving in the right direction, and I appreciate your dedication to refining the content.

Reviewer #2: The manuscript has undergone revisions, incorporating a modified title, “Transcriptional alterations of peanut root during interaction with growth-promoting Tsukamurella tyrosinosolvens strain P9”. Numerous comments have been addressed except a few.

The primary objective of the study was to explore the molecular response mechanism of peanut seedlings under the influence of strain P9 and to provide direct evidence for an in-depth analysis of the growth-promoting mechanism of strain P9 on peanuts. Their previous study reported that Tsukamurella tyrosinosolvens P9 exhibits strong phosphate solubilization, indole-3-acetic acid (IAA) secretion, and siderophore production. However, the corresponding genes were not extensively discussed except for the role of auxin in the revised manuscript. The root characteristics have a positive correlation with phosphate utilization and the phosphate solubilization also depends on the production of organic acids that helps in maintaining the pH of the environment for phosphate utilization. The modified version of the manuscript could be enhanced by addressing these issues, aligning with the authors' goal of elucidating the growth-promoting molecular mechanism of strain P9.

In response to this concern, the authors acknowledge that they did not delve deeply into the genes associated with the growth-promoting molecular mechanisms of Tsukamurella tyrosinosolvens P9. They appreciate the suggestion and express openness to further studying the growth-promoting molecular mechanisms of the P9 strain. However, the query arises about the significance of the present investigation if the molecular mechanism is not thoroughly addressed, considering the title "Transcriptional alterations of peanut root during interaction with growth promoting Tsukamurella tyrosinosolvens strain P9."

Moreover, on page 9, line 165, there is a question about the term "multiple metabolisc pathways," with a suggestion that it might be a typographical error, possibly intended to be "metabolic."

7. PLOS authors have the option to publish the peer review history of their article (what does this mean?). If published, this will include your full peer review and any attached files.

Reviewer #1: **Yes: **Rajarshi Sanyal

Reviewer #2: No

---

## [Author Response · Author response to Decision Letter 1]

18 Dec 2023

Dear Editor,

Please find attached a revised version of our manuscript “Transcriptional alterations of peanut root during interaction with growth-promoting Tsukamurella tyrosinosolvens strain P9”.

Thank you for your kind letter regarding our manuscript, and the referees’ remarks. We have revised and modified the MS in accordance with the every reviewers’ comments, and revisions in the text are shown using yellow highlight for additions, and strikethrough font for deletions. And we also responded point-by-point to reviewer comments as listed below. Accordingly, we changed the serial number of the Reference because of some reference’s addition.

We hope that the revisions in the manuscript and our accompanying responses will be sufficient to make our manuscript suitable for publication in PLOS ONE. We shall look forward to hearing from you at your earliest convenience.

Yours sincerely,

First author, Xue Bai; 

Corresponding author, Lizhen Han.

Corresponding author E-mail: lzhan1@gzu.edu.cn; hanlizhen11@163.com

Responses to the comments of Reviewer 1

I appreciate the effort the authors have put into incorporating the suggested changes. It's evident that the authors have taken the feedback seriously, and the manuscript is progressing well. However, I would like to draw your attention to sentence construction and grammatical errors.

1. There are still some grammatical errors and spelling mistakes throughout the manuscript (for example: Line 58, 182, 185, 192, 188, 205, 250, 332, 333, 364, 367, 377, 420, 466, 499, 505, 549, 558, 561, 634, 651, 656, 658, 665). The use of the phrase, “in regard to” has been used excessively, making the story monotonous. I suggest using a grammar checking software to resolve this.

Response:

Thanks for your suggestion. We are sorry about for that grammatical errors and spelling mistakes. We examined carefully and modified all these sections. And we changed “regarding”, “about”, “as for ” instead of “in regard to”, too.. 

2. Line 65: Please correct the spelling of peanut type. It should be“Virginia runner”.

Response:

Thanks, "Verginia runner type" has been changed to "Virginia runner type" in Line 61 of the revised manuscript.

3. The link given in data availability statement is not working as of 27.11.23. Overall, I believe the manuscript is moving in the right direction, and I appreciate your dedication to refining the content..

Response:

 I am sorry, an open access time was set when the database was uploaded, now the database has been opened with the link of https://www.ncbi.nlm.nih.gov/bioproject/PRJNA991079. And we modified this linkage in Line110. 

Responses to the comments of Reviewer 2

The manuscript has undergone revisions, incorporating a modified title, “Transcriptional alterations of peanut root during interaction with growth-promoting Tsukamurella tyrosinosolvens strain P9”. Numerous comments have been addressed except a few.

The primary objective of the study was to explore the molecular response mechanism of peanut seedlings under the influence of strain P9 and to provide direct evidence for an in-depth analysis of the growth-promoting mechanism of strain P9 on peanuts. Their previous study reported that Tsukamurella tyrosinosolvens P9 exhibits strong phosphate solubilization, indole-3-acetic acid (IAA) secretion, and siderophore production. However, the corresponding genes were not extensively discussed except for the role of auxin in the revised manuscript. The root characteristics have a positive correlation with phosphate utilization and the phosphate solubilization also depends on the production of organic acids that helps in maintaining the pH of the environment for phosphate utilization. The modified version of the manuscript could be enhanced by addressing these issues, aligning with the authors' goal of elucidating the growth-promoting molecular mechanism of strain P9.

In response to this concern, the authors acknowledge that they did not delve deeply into the genes associated with the growth-promoting molecular mechanisms of Tsukamurella tyrosinosolvens P9. They appreciate the suggestion and express openness to further studying the growth-promoting molecular mechanisms of the P9 strain. However, the query arises about the significance of the present investigation if the molecular mechanism is not thoroughly addressed, considering the title "Transcriptional alterations of peanut root during interaction with growth promoting Tsukamurella tyrosinosolvens strain P9."

Response:

Thanks for your suggestion. We added the paragraph in Line578-603 to associated changes in the transcriptome of peanut root with the growth-promoting properties of P9 in phosphorus solubilizing and siderophore secretion. And accordingly, we added cited articles in References. That is, “Our previous study confirmed that the strain P9 has a good ability to dissolve phosphorus and produce siderophore. Phosphorus is an essential element for the growth and development of plants. In order for plants to use insoluble phosphate in the environment, it must first be dissolved through the action of organic acids [Brito et al. 2020]. During the early stage of co-culture with strain P9, the PDC and ALDH genes, which catalyse the production of pyruvate and acetic acid, respectively, were up-regulated. Additionally, the LPIN, PDAT, phospholipase C and lysophospholipase II genes, which are involved in glycerophospholipid and glycerolipid metabolism, were also up-regulated. In the later stage, the genes encoding PDC and ADH was up-regulated to a greater extent (7.92-fold and 8.35-fold, respectively), and the genes encoding D-lactate dehydrogenase, malate dehydrogenase and pyruvate decarboxyase, which catalyse the production of lactic acid and malic acid, and the genes encoding choline-phosphate cytase and GPAT, which are involved in the synthesis of phospholipids and phosphoinositides, were significantly up-regulated. Our additional study also confirmed that organic acids are one of the main components of root extracts from peanut seedlings inoculated with the P9 strain [Jiang et al. 2023]. So, it was speculated that the P9 strain not only promoted the formation of organic acids in peanut roots, but also provided more soluble phosphorus for peanuts through its phosphorus solubilsing ability, which obviously promoted plant root development. In addition, iron is closely related to chlorophyll synthesis and photosynthesis, and the regulation of iron homeostasis greatly affects cell signal transduction, which in turn affects root growth [Kobayashi et al. 2019; Liang 2022], but the efficiency of iron utilization by plants is very low. Siderophore secreted by PGPR are an effective strategy for plants to absorb and utilize iron [Tsai and Schmidt 2017]. After inoculation with the P9 strain, the chlorophyll content of the leaves increased, root development was more vigorous, and the expression of photosynthesis-related proteins PetC and PetF was up-regulated. It was speculated that the siderophore secreted by P9 provided sufficient iron for the growth of peanut seedlings, promoting the development of peanut roots and an increase in chlorophyll content. This was consistent with the good plant growth-promoting ability of the P9 strain”.

Brito LF, López MG, Straube L, Passaglia LMP, Wendisch VF. Inorganic phosphate solubilization by rhizosphere bacterium Paenibacillus sonchi: gene expression and physiological functions. Front Microbiol. 2020;11:588605. https://doi.org/10.3389/fmicb.2020.588605

Kobayashi T, Nozoye T, Nishizawa NK. Iron transport and its regulation in plants. Free Radical Bio Med. 2019;133:11–20. https://doi.org/10.1016/j.freeradbiomed.2018.10.439

Liang G. Iron uptake, signaling, and sensing in plants. Plant Communications. 2022;3(5):100349. https://doi.org/10.1016/j.xplc.2022.100349

Tsai HH, Schmidt W. One way. Or another? Iron uptake in plants. New Phytologist. 2017;214(2):500–505. https://doi.org/10.1111/nph.14477

Moreover, on page 9, line 165, there is a question about the term "multiple metabolisc pathways," with a suggestion that it might be a typographical error, possibly intended to be "metabolic.".

Response:

Thanks, it’s our spelling mistakes. we changed "metabolic" instead of "metabolisc" in Line 170.

---

## [Decision Letter · Decision Letter 2]

2 Jan 2024

PONE-D-23-21487R2Transcriptional alterations of peanut root during interaction with growth-promoting  Tsukamurella tyrosinosolvens  strain P9PLOS ONE

Dear Dr. Lizhen,

Thank you for submitting your manuscript to PLOS ONE. After careful consideration, we feel that it has merit but does not fully meet PLOS ONE’s publication criteria as it currently stands. Therefore, we invite you to submit a revised version of the manuscript that addresses the points raised during the review process.

Reviewer 1 has suggested few minor corrections. Please make these corrections and submit the manuscript.

We look forward to receiving your revised manuscript.

Kind regards,

Anil Kumar Singh, Ph.D.

Academic Editor

PLOS ONE

Journal Requirements:

Reviewers' comments:

Reviewer's Responses to Questions

**Comments to the Author**

1. If the authors have adequately addressed your comments raised in a previous round of review and you feel that this manuscript is now acceptable for publication, you may indicate that here to bypass the “Comments to the Author” section, enter your conflict of interest statement in the “Confidential to Editor” section, and submit your "Accept" recommendation.

Reviewer #1: (No Response)

Reviewer #2: All comments have been addressed

2. Is the manuscript technically sound, and do the data support the conclusions?

Reviewer #1: Yes

Reviewer #2: Yes

3. Has the statistical analysis been performed appropriately and rigorously? 

Reviewer #1: Yes

Reviewer #2: Yes

4. Have the authors made all data underlying the findings in their manuscript fully available?

Reviewer #1: Yes

Reviewer #2: Yes

5. Is the manuscript presented in an intelligible fashion and written in standard English?

Reviewer #1: Yes

Reviewer #2: Yes

6. Review Comments to the Author

Reviewer #1: The modifications made to the manuscript, particularly in response to the previous comments have significantly improved the clarity and precision of the text. The provided link is now functional, addressing a previous issue. Overall, the study presents valuable insights; however, I have some specific comments and suggestions to enhance the clarity and precision of the manuscript.

Line 56-59: The statement "….to investigate their interaction" suggests that transcriptome analysis confirms the interaction, which may be misleading. Transcriptome analysis provides an overview resulting from the interaction, if any, but does not confirm it. I recommend modifying the sentence to accurately convey this point. Additionally, the remark, "...aimed to explore the molecular response mechanism," is somewhat ambiguous, as the study did not unveil a molecular mechanism per se. Rather, it focused on elucidating transcriptional alterations in peanut due to Tsukamurella tyrosinosolvens strain P9 inoculation. Therefore, I suggest refining the statements to align more closely with the study's actual findings.

Line 162: Please consider removing the word "were" to improve the sentence structure.

In my initial review, I mentioned the potential significance of the study's novelty. I would like to emphasize that if the authors can further highlight the significance of their study, which may not be adequately underscored in the current manuscript, it would substantially enhance the paper's potential for publication.

Reviewer #2: The manuscript has significantly improved as the authors have addressed the required revisions. The study was conducted on the molecular response mechanism of peanut seedlings to strain P9. They analyzed the transcript alterations to see the growth-promoting effects of strain P9 on peanuts. However, they could have enhanced the manuscript by incorporating a working model that elucidates the molecular mechanisms underlying the growth-promoting effects, providing a more comprehensive understanding of their claims.

7. PLOS authors have the option to publish the peer review history of their article (what does this mean?). If published, this will include your full peer review and any attached files.

Reviewer #1: **Yes: **Rajarshi Sanyal

Reviewer #2: No

---

## [Author Response · Author response to Decision Letter 2]

6 Jan 2024

Dear Editor,

Please find attached a revised version of our manuscript “Transcriptional alterations of peanut root during interaction with growth-promoting Tsukamurella tyrosinosolvens strain P9”.

Thank you for your kind letter regarding our manuscript, and the referees’ suggestions. We have revised and modified the MS in accordance with the every reviewers’ comments, and revisions in the text are shown using yellow highlight for additions, and strikethrough font for deletions. And we also responded point-by-point to reviewer comments as listed below. Accordingly, we added the cited references, and changed the serial number of the Reference because of some reference’s addition.

We hope that the revisions in the manuscript and our accompanying responses will be sufficient to make our manuscript suitable for publication in PLOS ONE. We shall look forward to hearing from you at your earliest convenience.

Yours sincerely,

First author, Xue Bai; 

Corresponding author, Lizhen Han.

Corresponding author E-mail: lzhan1@gzu.edu.cn; hanlizhen11@163.com

Responses to the comments of Reviewer 1

The modifications made to the manuscript, particularly in response to the previous comments have significantly improved the clarity and precision of the text. The provided link is now functional, addressing a previous issue. Overall, the study presents valuable insights; however, I have some specific comments and suggestions to enhance the clarity and precision of the manuscript.

1. Line 56-59: The statement "….to investigate their interaction" suggests that transcriptome analysis confirms the interaction, which may be misleading. Transcriptome analysis provides an overview resulting from the interaction, if any, but does not confirm it. I recommend modifying the sentence to accurately convey this point. Additionally, the remark, "...aimed to explore the molecular response mechanism," is somewhat ambiguous, as the study did not unveil a molecular mechanism per se. Rather, it focused on elucidating transcriptional alterations in peanut due to Tsukamurella tyrosinosolvens strain P9 inoculation. Therefore, I suggest refining the statements to align more closely with the study's actual findings.

Response:

Thanks for your suggestion. We changed “to study the gene expression of peanut roots after inoculation with the growth-promoting P9 strain” instead of “to investigate their interaction”. And in order to make our studying aim clearly, we modified this sentence, that is, “our aim was to provide direct evidence for the molecular mechanism of strain P9 in promoting peanut growth, and to provide a theoretical basis for using Tsukamurella tyrosinosolvens strain P9 to the field level”. 

2. Line 162: Please consider removing the word "were" to improve the sentence structure..

Response:

Thanks, we deleted this words.

3. In my initial review, I mentioned the potential significance of the study's novelty. I would like to emphasize that if the authors can further highlight the significance of their study, which may not be adequately underscored in the current manuscript, it would substantially enhance the paper's potential for publication..

Response:

 It’s my negligence, your suggestion made our studying more meaniful. Thanks. We added some studying progress about this rare actinobacteria, Tsukamurella tyrosinosolvens in Line 49-53, and highlighted the novelty of this strain in Line 55. 

Responses to the comments of Reviewer 2

The manuscript has significantly improved as the authors have addressed the required revisions. The study was conducted on the molecular response mechanism of peanut seedlings to strain P9. They analyzed the transcript alterations to see the growth-promoting effects of strain P9 on peanuts. However, they could have enhanced the manuscript by incorporating a working model that elucidates the molecular mechanisms underlying the growth-promoting effects, providing a more comprehensive understanding of their claims.

Response:

Thanks for your suggestion. In fact, the research on strain P9 has been ongoing. Since we isolated this strain of rare actinobacterium with strong phosphorus solubilization, Tsukamurella tyrosinosolvens P9, we found that this strain also has multiple growth-promoting properties such as secretion of IAA and siderophore, which can significantly promote the growth of peanut; while the underlying mechanism of its growth promotion is unclear, thus we tried to analyze it in a comprehensive way from different perspectives and aspects. It is well known that PGPR has a complex interaction with plants. In the previous study, we collected peanut root extraction (REs) and explored the effect of its addition on the transcriptome of P9 strain, especially on the expression of genes related to the growth-promoting function, and found that the peanut REs could significantly promote the growth and growth-promoting characteristics of P9 (Jiang et al. 2023). In this study, we established a "P9 strain-peanut co-culture system" and analyzed the transcriptome of peanut after inoculation with P9 strain, in order to explore how P9 strain promotes the growth of peanut from the perspective of gene expression. In this way, we can comprehensively assess the performance of P9 strain in the growth promotion function of peanut, so we believe that the paper is relatively comprehensive and complete.

Of course, your suggestion is very much appreciated, and we realise that if P9 and peanut in the co-culture system were studied simultaneously, the study would be more in-depth and more meaningful conclusions would be obtained. In fact, we are still conducting in-depth research on the growth-promoting mechanism of this strain; as P9 is a rare actinomycete, we are currently exploring and constructing a genetic transformation system of P9 strain to elucidate its growth-promoting properties on peanut growth; in addition, we have also found that P9 enhances drought resistance in peanut, and we are now studying using the co-culture system, hoping that we can more clearly analyze the mechanism of PGPR-plant interaction under adverse conditions.

---

## [Editor Report · Decision Letter 3]

23 Jan 2024

Transcriptional alterations of peanut root during interaction with growth-promoting  Tsukamurella tyrosinosolvens  strain P9

PONE-D-23-21487R3

Dear Dr. Lizhen,

We’re pleased to inform you that your manuscript has been judged scientifically suitable for publication and will be formally accepted for publication once it meets all outstanding technical requirements.

Kind regards,

Anil Kumar Singh, Ph.D.

Academic Editor

PLOS ONE
---

## [Editor Report · Acceptance letter]

7 Feb 2024

PONE-D-23-21487R3 

PLOS ONE

Dear Dr. Han, 

I'm pleased to inform you that your manuscript has been deemed suitable for publication in PLOS ONE. Congratulations! Your manuscript is now being handed over to our production team.

Kind regards, 

on behalf of

Dr. Anil Kumar Singh 

Academic Editor

PLOS ONE